# INVERSE CONTEXTUAL BANDITS:
## LEARNING HOW BEHAVIOR EVOLVES OVER TIME

### ABSTRACT

Understanding a decision-maker's priorities by observing their behavior is critical for transparency and accountability in decision processes—such as in healthcare. Though conventional approaches to policy learning almost invariably assume stationarity in behavior, this is hardly true in practice: Medical practice is constantly evolving as clinical professionals fine-tune their knowledge over time. For instance, as the medical community's understanding of organ transplantations has progressed over the years, a pertinent question is: How have actual organ allocation policies been evolving? To give an answer, we desire a policy learning method that provides *interpretable* representations of decision-making, in particular capturing an agent's *non-stationary* knowledge of the world, as well as operating in an *offline* manner. First, we model the evolving behavior of decision-makers in terms of contextual bandits, and formalize the problem of *Inverse Contextual Bandits* (ICB). Second, we propose two concrete algorithms as solutions, learning parametric and nonparametric representations of an agent's behavior. Finally, using both real and simulated data for liver transplantations, we illustrate the applicability and explainability of our method, as well as benchmarking and validating the accuracy of our algorithms.

## 1 INTRODUCTION

Modeling decision-making policies is a central concern in computational and behavioral science, with key applications in healthcare [1], economics [2], and cognition [3]. The business of *policy learning* is to determine an agent's decision-making policy given observations of their behavior. Typically, the objective is either to replicate the behavior of some demonstrator (cf. "imitation learning") [4, 5], or to match their performance on the basis of some reward function (cf. "apprenticeship learning") [6, 7]. However, equally important is the pursuit of "descriptive modeling" [8, 9]—that is, of learning interpretable parameterizations of behavior for auditing, quantifying, and understanding decision-making policies. For instance, recent work has studied representing behaviors in terms of rules [10], goals [11], intentions [12], preferences [13], subjective dynamics [14], as well as counterfactual outcomes [15].

**Evolving Behaviors** In this work, we ask a novel descriptive question: *How has the observed behavior changed over time?* While conventional approaches to policy learning almost invariably assume that decision-making agents are stationary, this is rarely the case in practice: In many settings, behaviors evolve constantly as decision-makers learn more about their environment and adjust their policies accordingly. In fact, disseminating new knowledge from medical research into actual clinical practice is itself a major endeavor in healthcare [16–18]. This research question is new: While capturing "variation in practice" in observed data has been studied in the context of demonstrations containing mixed policies [19, 20], multiple tasks [21, 22], and subgroup-/patient-specific preferences [8], little work has attempted to capture variation in practice *over time* as an agent's knowledge of the world evolves.

**Example (Organ Allocations)** As our core application, consider organ allocation practices for liver transplantations: In the medical community, our understanding of organ transplantations has changed numerous times over past decades [23–25]. Thus an immediate question lies in *how* actual organ allocation practices have changed over the years. Having a data-driven, quantitative, and—importantly—interpretable description of how practices have evolved is crucial: It would enable policy-makers to evaluate if the policies they introduced have had the intended impact on practice; this would in turn play a substantial role in designing better policies going forward [8] (see Figure 1).

To tackle questions of this form, we desire a policy learning method that satisfies three key desiderata:

- It should provide an (i.) *interpretable* description of observed behavior (see Appendix C).

- It should be able to capture an agent's (ii.) *non-stationary* knowledge of the world.

- It should operate in an (iii.) *offline* manner—since online experimentation is impossible in high-stakes environments such as healthcare.

**Inverse Contextual Bandits** To accomplish this, we first identify the organ allocation problem as a *contextual bandits* problem [26–28]: Given each arriving instance of patient and organ features (i.e. the context), an agent makes an allocation decision (i.e. the action), whence some measure of feedback is perceived (i.e. internal reward). Crucially, the environment (i.e. precisely, its reward dynamics) is unknown to the agent and must be *actively learned*, so the agent maintains beliefs about their environment (i.e. internal knowledge). Not only must an agent select actions that "exploit" the knowledge they have, but they must also select actions that "explore" the environment to update their knowledge. Thus the behavior of a learning agent is naturally modeled as a generalized bandit strategy—that is, how to take actions based on their knowledge, and how to update their knowledge based on the outcomes of their actions.

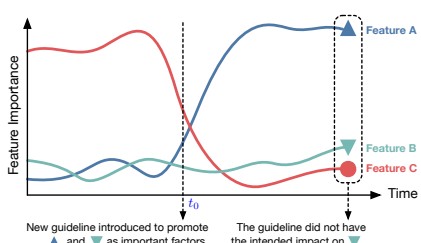

*Figure 1. Evaluating Impact of Policies via ICB. Suppose that policy-makers introduce a new policy to promote Features A and B as main factors of consideration in decision-making (whereas clinicians rely on Feature C at the time), and the policy succeeds at establishing Feature A as such but fails to do so for Feature B. Using ICB, we can infer the time-varying importances of these features and observe quantitatively that importance of Feature B stayed the same despite the intentions of the new policy. Having made this observation, the policy-makers can now update their policy accordingly.*

Now, the *forward* contextual bandits problem asks the (normative) question: Given an unknown environment, what is an effective bandit strategy that minimizes some notion of regret? By contrast, our focus here is instead on the opposite direction—that is, in formalizing and solving the problem of *Inverse Contextual Bandits* ("ICB"): We ask the (descriptive) question: Given demonstrated behavior from a decision-making agent, how has the agent's knowledge been evolving over time? Precisely, we wish to learn interpretable representations of generalized bandit strategies from the observable context-action pairs generated by those strategies—regardless of whether those strategies are effective.

**Contributions** Our contributions are three-fold. In the sequel, we first formalize the ICB problem, identifying it with the data-driven objective of inferring an agent's internal reward function along with their internal trajectory of beliefs about the environment (Section 2). Second, we propose two concrete learning algorithms, imposing different specifications regarding the agent's behavioral strategy: The first parameterizes the agent's knowledge in terms of Bayesian updates, whereas the second makes the milder specification that the agent's behavior evolves smoothly over time (Section 3). Third, through both simulated and real-world data for liver transplantations, we illustrate how ICB can be applied as an investigative device for recovering and explaining the evolution of organ allocation practices over the years, as well as benchmarking and validating the accuracy of our algorithms (Section 5).

## 2 INVERSE CONTEXTUAL BANDITS

**Preliminaries** Consider a *Markov decision process* of the form $\mathbb{D} := (X, A, \mathcal{R}, \mathcal{T})$, where $S$ indicates the state space, $A$ the action space, $\mathcal{R} \in \Delta(\mathbb{R})^{X \times A}$ the reward dynamics, and $\mathcal{T} \in \Delta(X)^{X \times A}$ the transition dynamics. At each time $t \in \mathbb{Z}_+$, the decision-making agent is presented with some state $x_t \in X$ and decides to take an action $a_t \in A$, whence an immediate reward $r_t \sim \mathcal{R}(x_t, a_t)$ is received, and a subsequent state $s_{t+1} \sim \mathcal{T}(x_t, a_t)$ is presented. Let the *decision-making policy* employed by the agent be denoted $\pi \in \Delta(A)^X$, such that actions at any time are sampled according to $a_t \sim \pi(x_t)$.

In the forward direction, given dynamics $\mathcal{R}, \mathcal{T}$, the *reinforcement learning* problem ("RL") deals with determining the optimal policy that maximizes some notion of expected cumulative rewards [29]: $\pi^*_{\mathcal{R},\mathcal{T}} := \operatorname{argmax}_{\pi \in \Delta(A)^X} \mathbb{E}_{\pi,\mathcal{R},\mathcal{T}} \sum_t r_t$. In the opposite direction, given some observed behavior policy $\pi_b$ and transition dynamics $\mathcal{T}$, the *inverse reinforcement learning* problem ("IRL") deals with determining the reward dynamics $\mathcal{R}$ with respect to which $\pi_b$ appears optimal. For instance, the classic max-margin approach seeks [30]: $\mathcal{R}^* := \operatorname{argmin}_{\mathcal{R}}(\max_\pi \mathbb{E}_{\pi,\mathcal{R},\mathcal{T}} \sum_t r_t - \mathbb{E}_{\pi_b,\mathcal{R},\mathcal{T}} \sum_t r_t)$.

**Having Dynamics Access to Having No Dynamics Access** In conventional RL (and IRL), the decision-maker has unrestricted access to environment dynamics—either explicitly (i.e. $\mathcal{R}, \mathcal{T}$ are simply *known* and used in computing the optimal policy), or implicitly (i.e. the agent may *interact* freely with the environment during training). In contrast, in our setting the agent has no such

luxuries—not only are the dynamics not known to the agent, but neither do they enjoy a distinct sandboxed training phase prior to live deployment. Without such access, the agent must consider both the information they may gain when taking actions (cf. "exploration") and also the expected rewards due to those actions (cf. "exploitation"). Note that this property results in much more difficult learning problems—in both the forward and inverse directions (see Appendix B for a more detailed discussion).

Consider environments parameterized by *environment parameters* $\rho \in P$, such that the reward dynamics of an environment are given by $\mathcal{R}_\rho$, and the transition dynamics by $\mathcal{T}_\rho$. Let $\rho_{\text{env}}$ denote the true environment parameter, such that the actual rewards received by an agent are distributed as $r_t \sim \mathcal{R}_{\rho_{\text{env}}}(x_t, a_t)$, and the actual states encountered by the agent are distributed as $x_{t+1} \sim \mathcal{T}_{\rho_{\text{env}}}(x_t, a_t)$. Since $\rho_{\text{env}}$ is unknown, an agent takes actions on the basis of their beliefs about it; these beliefs are described by probability distributions $\mathcal{P}_\beta \in \Delta(P)$ parameterized by *belief parameters* $\beta \in B$. For each time $t$, let $\beta_t$ capture the agent's belief at the beginning of that step.

Each time step, the agent first samples an environment parameter $\rho_t \sim \mathcal{P}_{\beta_t}$ according to their belief $\beta_t$, then takes an action according to $\pi^*_{\mathcal{R}_{\rho_t}, \mathcal{T}_{\rho_t}}$. Note that this ensures that each action is taken with probability proportional to the probability with which the agent believes it to be optimal at time step $t$. Essentially, the agent's policy at time step $t$ is induced by their belief $\beta_t$: $\pi_{\beta_t}(x)[a] := \mathbb{E}_{\rho \sim \mathcal{P}_{\beta_t}}[\pi^*_{\mathcal{R}_\rho, \mathcal{T}_\rho}(x)[a]]$. After receiving a reward $r_t$, the agent updates their current belief parameter according to a (possibly-stochastic) *belief-update function* $f \in \Delta(B)^{B \times X \times A \times \mathbb{R}}$, such that $\beta_{t+1} \sim f(\beta_t, x_t, a_t, r_t)$. Together with the initial belief parameter $\beta_1$, the agent's $t$-th step belief is a (possibly-stochastic) function of the history $h_t = \{\boldsymbol{x}_{1:t-1}, \boldsymbol{a}_{1:t-1}, \boldsymbol{r}_{1:t-1}\}$ defined recursively via $f$.

## 2.1 CONTEXTUAL BANDITS SETTING

In this work, we consider state transitions that occur independently of past states and actions. Due to this property, decisions can be made greedily without consideration of what future states may be, and yields a contextual bandits problem [26–28]. Note that this captures the organ allocation problem well: Distributions of newly arriving organs are largely independent of prior allocation decisions; in fact, transplantation and allocation policies are often modeled in bandit-like settings [31, 32]. Formally:

**Definition 1 (Contextual Bandits)** Consider a decision-making problem $\mathbb{D} := (X, A, \mathcal{R}, \mathcal{T})$, where $\mathcal{R}, \mathcal{T}$ are unknown to the agent. Let $\mathcal{T}(x, a) = \mathcal{T}'$ for some $\mathcal{T}' \in \Delta(X)$, for all $x \in X, a \in A$, such that policies are greedy:

$$\text{supp}(\pi^*_t(x_t)) = \text{argmax}_{a_t \in A} \bar{\mathcal{R}}_{\rho_t}(x_t, a_t) , \tag{1}$$

where $\bar{\mathcal{R}}_\rho(x, a) := \mathbb{E}_{r \sim \mathcal{R}_\rho(x,a)}[r]$ indicates the mean reward function, and ties are broken arbitrarily. Given a space of environment parameterizations $P$, the *contextual bandits* problem is to design a space of belief parameterizations $B$, and to determine the optimal belief-update function

$$f^* := \text{argmax}_{f \in \Delta(B)^{B \times X \times A \times \mathbb{R}}} \sum_t \mathbb{E}_{f, \pi_{\beta_t}, \mathcal{R}, \mathcal{T}}[r_t] . \tag{2}$$

Now, suppose that an agent follows a bandit-type policy for $T$ time steps; this would generate an *observational dataset* of contexts and actions $\mathcal{D} := \{\boldsymbol{x}_{1:T}, \boldsymbol{a}_{1:T}\}$ (adhering to the contextual bandits literature, we refer to states as "contexts" hereafter).[1] But since rewards $r_t$ and beliefs $\beta_t$ are quantities *internal* to the decision-maker, we assume that they are not part of the observational dataset: The former represents the agent's preferences over outcomes after observing contexts and taking actions, which is not explicitly observable; likewise, the latter represents the agent's beliefs about what kinds of outcomes their actions result in, which is also not explicitly observable. Thus we ask the novel question:

*From $\mathcal{D}$, can we infer the true environment parameter $\boldsymbol{\rho}_{env}$, as well as the belief trajectory $\boldsymbol{\beta}_{1:T}$?*

**Definition 2 (Inverse Contextual Bandits)** Consider again a contextual bandit problem $\mathbb{D} := (X, A, \mathcal{R}, \mathcal{T})$, and recall that dynamics $\mathcal{R}, \mathcal{T}$ are unknown to the agent. Given an observational dataset $\mathcal{D}$, and a family of environment parameterizations $P$ and belief parameterizations $B$, the *inverse contextual bandits* problem ("ICB") is to determine the true environment parameter $\rho_{\text{env}}$ and the belief parameters $\boldsymbol{\beta}_{1:T}$ (see Figure 2).

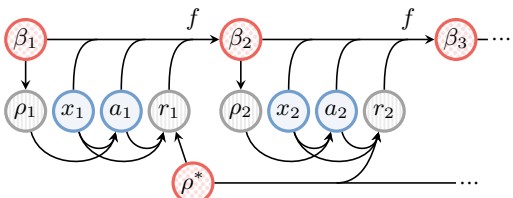

*Figure 2. Graphical Model for ICB.* We aim to infer the (dotted) red quantities given the (solid) blue ones.

---

[1] We also use "$x \in X$" to denote contexts (standard in contextual bandits) instead of "$s \in S$" (common in RL).

It is important to observe that the novelty here is *general*—that is, in posing the "inverse" question of how knowledge evolves over time. Focusing our attention on contextual bandits simply represents a specific choice of problem setting: In this first work on tackling the *inverse* problem in modeling non-stationary agents, it presents a more tractable problem for analysis, and is moreover especially suited for our motivating application in organ allocations. Note that in the bandit setting, the fact that transition dynamics are unknown to the agent is ultimately inconsequential due to greedy policies; we refer to $\rho$ simply as "reward parameter" hereafter. We leave generalization to arbitrary transition dynamics $\mathcal{T} \in \Delta(X)^{X \times A}$ to future work. Also note that ICB itself is not a bandit problem (see Appendix C).

**Remark 1 (Environment vs. State Beliefs)** When speaking of inferring "beliefs" here in our setting, we are referring to *environment beliefs*—that is, an agent's knowledge of the environment's rewards. It is important to distinguish this from *state beliefs* in partially-observed environments [33, 34]—that is, an agent's estimate of which latent state they are in. State beliefs are computed by an agent using their (known) environment parameters, whereas environment beliefs concern the environment's (unknown) parameters themselves. State beliefs can be easily inferred [35]: all factors that contribute to state-belief updates (i.e. observations and actions) are observable; on the other hand, environment beliefs are more technically challenging due to latent factors (i.e. internal rewards $\boldsymbol{r}_{1:T}$) that are never observable. Because of the same reason, tackling ICB with conventional IO-HMM inference methods is not possible either, even though the agent can be viewed as an IO-HMM (see Appendix C).

**Remark 2 (Subjective vs. Objective Reward)** When speaking of learning what "rewards" an agent is optimizing, the novelty here that our objective refers to the full *belief trajectory* $\boldsymbol{\beta}_{1:T}$ of the agent. It is important to distinguish this from simply learning the *ground-truth parameter* $\rho_{\text{env}}$ as in typical IRL. The ground-truth parameter is an objective quantity capturing the "prescriptive" notion of what the agent *ought to be* optimizing, whereas the belief trajectory is a subjective quantity capturing the "descriptive" notion of what the agent *appears to be* optimizing. Existing work in learning from non-stationary agents has focused solely on $\rho_{\text{env}}$, which is all that is required for apprenticeship [36, 37]. In contrast, for the goal of understanding behavior (and how it evolves), Definition 2 brings $\boldsymbol{\beta}_{1:T}$ to focus.

## 3 BAYESIAN INVERSE CONTEXTUAL BANDITS

We operate in the standard contextual bandits setting [26]: Consider a $d$-dimensional action space $A$, and suppose at each time $t$ the agent observes a $k$-dimensional context $x_t[a] \in \mathbb{R}^k$ for each possible action, thus the context space is given by $X := \mathbb{R}^{d \times k}$. Let the space of reward parameters be $P := \mathbb{R}^k$. Rewards are assumed linear with respect to contexts $x_t$, and normally-distributed with mean $\bar{\mathcal{R}}_\rho(x, a) = \langle \rho, x[a] \rangle$ and known variance $\sigma^2$; that is, $\mathcal{R}_\rho(x, a) := \mathcal{N}(\langle \rho, x[a] \rangle, \sigma^2)$ with $\langle \cdot, \cdot \rangle$ denoting the inner product. (Note that these assumptions can be relaxed later in Section 3.1 below).

**Belief Updates** Before tackling a much more general case, we begin by modeling the agent's beliefs as Gaussian posteriors over $\rho_{\text{env}}$, and their belief-update function $f$ as Bayesian updates. Formally, beliefs are described by the family of distributions $\mathcal{P}_\beta := \mathcal{N}(\mu, \Sigma)$, where $\beta := \{\mu, \Sigma\} \in \mathbb{R}^k \times \mathbb{R}^{k \times k}$, and $\mathcal{N}(\mu, \Sigma)$ is the multivariate Gaussian distribution with mean vector $\mu$ and covariance matrix $\Sigma$. At each time $t$, given the posterior $\beta_t = \{\mu_t, \Sigma_t\}$—such that $\rho_{\text{env}}|h_t \sim \mathcal{N}(\mu_t, \Sigma_t)$—and the observation $(x_t, a_t, r_t)$, the belief-update $f(\beta_t, x_t, a_t, r_t)$ is the Dirac delta centered at $\beta_{t+1} = \{\mu_{t+1}, \Sigma_{t+1}\}$, with:

$$\mu_{t+1} := \Sigma_{t+1}\left(\Sigma_t^{-1}\mu_t + \frac{1}{\sigma^2}r_t x_t[a_t]\right), \quad \Sigma_{t+1} := \left(\Sigma_t^{-1} + \frac{1}{\sigma^2}x_t[a_t]x_t[a_t]^\mathsf{T}\right)^{-1}, \quad (3)$$

Note that the initial belief $\beta_1 = \{\mu_1, \Sigma_1\}$ represents the agent's (unknown) prior over $\rho_{\text{env}}$. Finally, to facilitate Bayesian analysis, we consider soft-optimal policies as usual (see e.g. [38]). Formally, instead of picking actions uniformly from $\text{argmax}_{as \in A} \bar{\mathcal{R}}_{\rho_t}(x_t, a_t)$, they are chosen as follows, with $\alpha \in \mathbb{R}_+$ adjusting the stochasticity of action selection ($\alpha \to \infty$ recovers the hard-optimal case):

$$\pi^*_{\mathcal{R}_{\rho_t}}(x_t)[a_t] := e^{\alpha \bar{\mathcal{R}}_{\rho_t}(x_t, a_t)} / \sum_{a \in A} e^{\alpha \bar{\mathcal{R}}_{\rho_t}(x_t, a)} . \quad (4)$$

**Bayesian Learning** Under this setup, we propose an expectation-maximization (EM)-like algorithm that estimates the true reward parameter $\rho_{\text{env}}$ and the initial belief parameter $\beta_1$, and samples the belief trajectory $\boldsymbol{\beta}_{2:T}$ in alternating steps. First, the joint distribution of all quantities of interest is:

$$\mathbb{P}(\rho_{\text{env}}, \beta_1, \boldsymbol{r}_{1:T}, \boldsymbol{\rho}_{1:T}, \mathcal{D}) = \underbrace{\mathbb{P}(\rho_{\text{env}}, \beta_1)}_{\text{Prior}} \prod_{t=1}^{T} \underbrace{\mathbb{P}(x_t)}_{\mathcal{T}[x_t]} \underbrace{\mathbb{P}(\rho_t|\beta_t)}_{\mathcal{P}_{\beta_t}[\rho_t]} \underbrace{\mathbb{P}(a_t|x_t, \rho_t)}_{\pi^*_{\mathcal{R}_{\rho_t}}(x_t)[a_t]} \underbrace{\mathbb{P}(r_t|\rho_{\text{env}}, x_t, a_t)}_{\mathcal{R}_{\rho_{\text{env}}}(x_t, a_t)[r_t]} . \quad (5)$$

Note that since each belief parameter $\beta_t$ is a deterministic function of the initial belief parameter $\beta_1$ and the history $h_t$ (cf. Equation 3), rewards $r_{1:T-1}$ and belief parameters $\beta_{2:T}$ are essentially interchangeable given the initial belief and context-action pairs $\mathcal{D}$ (i.e. one is completely identified by the other). Then, starting with some initial estimates $\hat{\rho}_{\text{env}}$ and $\hat{\beta}_1$ for the true reward parameter and the initial belief parameter, we iteratively obtain better estimates by performing the following two steps:

- *Expectation step:* Compute the expected log-likelihood of $\rho_{\text{env}}, \beta_1$ given current estimates $\hat{\rho}_{\text{env}}, \hat{\beta}_1$:

$$\mathcal{Q}(\rho_{\text{env}}, \beta_1; \hat{\rho}_{\text{env}}, \hat{\beta}_1) := \mathbb{E}_{r_{1:T}, \boldsymbol{\rho}_{1:T} | \hat{\rho}_{\text{env}}, \hat{\beta}_1, \mathcal{D}}[\log \mathbb{P}(r_{1:T}, \boldsymbol{\rho}_{1:T}, \mathcal{D} | \rho_{\text{env}}, \beta_1)] , \qquad (6)$$

which we shall approximate by sampling reward values and reward parameters $\{r_{1:T}^{(i)}, \boldsymbol{\rho}_{1:T}^{(i)}\}_{i=1}^N$ from the distribution $\mathbb{P}(r_{1:T}, \boldsymbol{\rho}_{1:T} | \hat{\rho}_{\text{env}}, \hat{\beta}_1, \mathcal{D})$ using Markov chain Monte Carlo (MCMC) methods.

- *Maximization step:* Compute new estimates $\hat{\rho}'_{\text{env}}, \hat{\beta}'_1$ that yield an improved expected log-posterior:

$$\mathcal{Q}(\hat{\rho}'_{\text{env}}, \hat{\beta}'_1; \hat{\rho}_{\text{env}}, \hat{\beta}_1) + \log \mathbb{P}(\hat{\rho}'_{\text{env}}, \hat{\beta}'_1) > \mathcal{Q}(\hat{\rho}_{\text{env}}, \hat{\beta}_1; \hat{\rho}_{\text{env}}, \hat{\beta}_1) + \log \mathbb{P}(\hat{\rho}_{\text{env}}, \hat{\beta}_1) , \qquad (7)$$

which can be achieved using gradient-based methods.

**Sampling Reward Values and Reward Parameters** Observe that reward values $r_{1:T}$ are Gaussian-distributed when conditioned on reward parameters $\boldsymbol{\rho}_{1:T}$ (which means they can be sampled exactly):

**Lemma 1** Let $r_{1:T} = [r_1 \cdots r_T] \in \mathbb{R}^T$ and let $X_t = (1/\sigma^2)[x_1[a_1] \cdots x_t[a_t] \; \mathbf{0}_{k,(T-t)}] \in \mathbb{R}^{k \times T}$, where $\mathbf{0}_{i,j}$ is the $i$-by-$j$ zero matrix. Then we have that $r_{1:T} | \boldsymbol{\rho}_{1:T}, \hat{\rho}_{\text{env}}, \hat{\beta}_1, \mathcal{D} \sim \mathcal{N}(\tilde{\mu}, \tilde{\Sigma})$, where

$$\tilde{\mu} := \tilde{\Sigma}\left(X_T^\mathsf{T}\hat{\rho}_{\text{env}} + \textstyle\sum_{t=2}^T X_{t-1}^\mathsf{T}(\rho_t - \hat{\Sigma}_t \hat{\Sigma}_1^{-1} \hat{\mu}_1)\right), \quad \tilde{\Sigma} := \left(\frac{1}{\sigma^2}I + \textstyle\sum_{t=2}^T X_{t-1}^\mathsf{T} \hat{\Sigma}_t X_{t-1}\right)^{-1} . \quad (8)$$

*Proof.* All proofs are found in Appendix D. $\qquad \square$

This motivates a Gibbs-like sampling procedure whereby samples from $\mathbb{P}(r_{1:T}, \boldsymbol{\rho}_{1:T} | \hat{\rho}_{\text{env}}, \hat{\beta}_1, \mathcal{D})$ are approximated by sampling $r_{1:T}$ and $\boldsymbol{\rho}_{1:T}$ in an alternating fashion from their respective conditional distributions $\mathbb{P}(r_{1:T} | \boldsymbol{\rho}_{1:T}, \hat{\rho}_{\text{env}}, \hat{\beta}_1, \mathcal{D})$ and $\mathbb{P}(\boldsymbol{\rho}_{1:T} | r_{1:T}, \hat{\rho}_{\text{env}}, \hat{\beta}_1, \mathcal{D})$ instead. However, note that sampling reward parameters $\boldsymbol{\rho}_{1:T}$ from its conditional is not as easy to perform exactly as sampling reward values $r_{1:T}$. We achieve it by first observing that $\rho_t$'s are independent from each other conditioned on $r_{1:T}$:

$$\begin{aligned} \mathbb{P}(\boldsymbol{\rho}_{1:T} | r_{1:T}, \hat{\rho}_{\text{env}}, \hat{\beta}_1, \mathcal{D}) \\ \propto \textstyle\prod_{t=1}^T \mathbb{P}(\rho_t | \hat{\beta}_t) \mathbb{P}(a_t | x_t, \rho_t) . \end{aligned} \qquad (9)$$

Then, we sample each $\rho_t$ in turn by performing a single iteration of the Metropolis-Hastings algorithm using $\mathcal{P}_{\hat{\beta}_t}$ as the proposal distribution. Algorithm 1 (Bayesian ICB) summarizes the overall procedure.

---

**Algorithm 1** Bayesian ICB

1: **Parameters:** variance of rewards $\sigma^2$, stochasticity of action selections $\alpha$, learning rate $\eta \in \mathbb{R}_+$
2: **Input:** dataset $\mathcal{D}$, prior $\mathbb{P}(\rho_{\text{env}}, \beta_1)$
3: Initialize $\hat{\rho}_{\text{env}}, \hat{\beta}_1 \sim \mathbb{P}(\rho_{\text{env}}, \beta_1)$
4: **loop**
5:     Initialize $r_{1:T}^{(0)}, \boldsymbol{\rho}_{1:T}^{(0)}$
6:     **for** $i \in \{1, 2, \ldots, N\}$ **do**
7:         Sample $r_{1:T}^{(i)} \sim \mathbb{P}(r_{1:T} | \boldsymbol{\rho}_{1:T}^{(i-1)}, \hat{\rho}_{\text{env}}, \hat{\beta}_1, \mathcal{D})$ ⌐ using Lemma 1
8:         **for** $t \in \{1, 2, \ldots, T\}$ **do**
9:             Sample $\rho', \rho'' \sim \mathcal{P}_{\beta_t[\hat{\beta}_1, x_{1:t-1}, a_{1:t-1}, r_{1:t-1}^{(i)}]}$
10:            $p \leftarrow \min\{1, \frac{\mathbb{P}(a_t | x_t, \rho_t = \rho')}{\mathbb{P}(a_t | x_t, \rho_t = \rho'')}\}$
11:            $\rho_t^{(i)} \leftarrow \rho'$ w.p. $p$, and $\rho''$ otherwise
12:         **end for**
13:     **end for**
14:     $\bar{\mathcal{Q}}(\rho_{\text{env}}, \beta_1) = \frac{1}{N}\sum_{i=1}^N \log \mathbb{P}(r_{1:T}^{(i)}, \boldsymbol{\rho}_{1:T}^{(i)}, \mathcal{D} | \rho_{\text{env}}, \beta_1)$
15:     $\{\hat{\rho}_{\text{env}}, \hat{\beta}_1\} \leftarrow \{\hat{\rho}_{\text{env}}, \hat{\beta}_1\}$
        $+ \eta[\nabla_{\{\rho_{\text{env}}, \beta_1\}} \bar{\mathcal{Q}}(\rho_{\text{env}}, \beta_1)]_{\{\rho_{\text{env}}, \beta_1\} = \{\hat{\rho}_{\text{env}}, \hat{\beta}_1\}}$
        $+ \eta[\nabla_{\{\rho_{\text{env}}, \beta_1\}} \log \mathbb{P}(\rho_{\text{env}}, \beta_1)]_{\{\rho_{\text{env}}, \beta_1\} = \{\hat{\rho}_{\text{env}}, \hat{\beta}_1\}}$
16: **end loop**
17: **Output:** $\hat{\rho}_{\text{env}}, \{\boldsymbol{\beta}_{1:T}^{(i)}[\hat{\beta}_1, x_{1:T}, a_{1:T}, r_{1:T}^{(i)}]\}_{i=1}^N$

---

### 3.1 Nonparametric Bayesian Inverse Contextual Bandits

So far, we have modeled the learning procedure of the agent with Bayesian updates, and estimated the parameters that characterize their learning procedure (i.e. the true reward parameter $\rho_{\text{env}}$ and the initial belief parameter $\beta_1$). Clearly, this approach can be similarly applied to different parameterizations of the agent's behavior (e.g. using a UCB-based model, or a general function approximator for $f$). In this section, instead of imposing a particular type of belief-update, we take a nonparametric approach and regard the agent's beliefs $\boldsymbol{\beta}_{1:T}$ more generally as a random process. First, we establish a prior $\mathbb{P}(\boldsymbol{\beta}_{1:T})$ over that belief process, then describe a procedure to sample from the posterior $\mathbb{P}(\boldsymbol{\beta}_{1:T} | \mathcal{D})$.

**Gaussian Process Prior** Different from above, now let the agent's beliefs be described by the family of distributions $\mathcal{P}_\beta := \mathcal{N}(\beta, \Sigma_P)$ for $\beta \in \mathbb{R}^k$, where $\Sigma_P \in \mathbb{R}^{k \times k}$ controls the variability of reward parameters sampled from beliefs. Let $\boldsymbol{\beta}_{1:T} = [\beta_1 \cdots \beta_T] \in \mathbb{R}^{k \times T}$; here we consider a multivariate Gaussian process as prior over belief parameters $\boldsymbol{\beta}_{1:T}$:

$$\text{vec}(\boldsymbol{\beta}_{1:T}) \sim \mathcal{N}(\mathbf{0}, \Sigma_T \otimes \Sigma_B) , \qquad (10)$$

where $\otimes$ denotes the Kronecker product, $\Sigma_T \in \mathbb{R}^{T \times T}$ controls the covariances between different time steps, and $\Sigma_B \in \mathbb{R}^{k \times k}$ controls the covariances between different components of a given belief $\beta_t$. Although our methodology is applicable for any arbitrary $\Sigma_T$, we shall fix $(\Sigma_T)_{ij} = \min\{i, j\}$ so our prior becomes a multivariate Wiener process, and differences $\delta_t := \beta_t - \beta_{t-1}$ between consecutive beliefs are independent and identically distributed according to $\mathcal{N}(0, \Sigma_B)$—that is $\mathbb{P}(\delta_t) \propto \exp(-\frac{1}{2}\delta_T^\mathsf{T}\Sigma_B^{-1}\delta_t)$. Intuitively, this means our prior favors belief trajectories that vary smoothly over time, where differences between consecutive beliefs are probabilistically bounded by $\Sigma_B$ (since larger changes in consecutive $\beta_t$'s are exponentially less likely in relation to $\Sigma_B$).

**Sampling Beliefs from the Posterior**  Having established a Gaussian process prior $\mathbb{P}(\beta_{1:T})$, observe that the posterior for $\beta_{1:T}$ is still a Gaussian process when conditioned on the reward parameters $\rho_{1:T}$:

**Lemma 2**  Let $\rho_{1:T} = [\rho_1 \cdots \rho_T] \in \mathbb{R}^{k \times T}$. Then we have that $\mathrm{vec}(\beta_{1:T})|\rho_{1:T}, \mathcal{D} \sim \mathcal{N}(\tilde{\mu}, \tilde{\Sigma})$, with

$$\tilde{\mu} := \tilde{\Sigma}(I \otimes \Sigma_P)^{-1}\,\mathrm{vec}(\rho_{1:T}), \quad \tilde{\Sigma} := \left((\Sigma_T \otimes \Sigma_B)^{-1} + (I \otimes \Sigma_P)^{-1}\right)^{-1}. \qquad (11)$$

Therefore, similar to the parametric version of our algorithm, we can approximate samples from the posterior $\mathbb{P}(\beta_{1:T}|\mathcal{D})$ by sampling $\beta_{1:T}$ and $\rho_{1:T}$ in an alternating fashion from their respective conditional distributions $\mathbb{P}(\beta_{1:T}|\rho_{1:T}, \mathcal{D})$ and $\mathbb{P}(\rho_{1:T}|\beta_{1:T}, \mathcal{D})$, and discarding the samples for $\rho_{1:T}$. Likewise, we sample each $\rho_t$ in turn by performing a single iteration of the Metropolis-Hastings algorithm. Algorithm 2 (Nonparametric Bayesian ICB) summarizes the overall sampling procedure. See Appendix C for a discussion of differences between Algorithm 1 and 2.

---

**Algorithm 2** Nonparametric Bayesian ICB

1: **Parameters:** covariance matrices $\Sigma_P$ and $\Sigma_B$
2: **Input:** dataset $\mathcal{D}$
3: Initialize $\rho_{1:T}^{(0)}, \beta_{1:T}^{(0)}$
4: **for** $i \in \{1, 2, \ldots, N\}$ **do**
5:    Sample $\beta_{1:T}^{(i)} \sim \mathbb{P}(\beta_{1:T}|\rho_{1:T}^{(i-1)}, \mathcal{D})$ ⌐ using Lemma 2
6:    **for** $t \in \{1, 2, \ldots, T\}$ **do**
7:       Sample $\rho', \rho'' \sim \mathcal{P}_{\beta_t^{(i)}}$
8:       $p \leftarrow \min\{1, \frac{\mathbb{P}(a_t|x_t, \rho_t=\rho')}{\mathbb{P}(a_t|x_t, \rho_t=\rho'')}\}$
9:       $\rho_t^{(i)} \leftarrow \rho'$ w.p. $p$, and $\rho''$ otherwise
10:   **end for**
11: **end for**
12: **Output:** $\{\beta_{1:T}^{(i)}\}_{i=1}^{N}$

---

## 4 RELATED WORK

**Policy Learning**  In modeling an agent's decision-making from their observed behavior, the overarching goal is often in *imitation learning* (i.e. to replicate their actions) or in *apprenticeship learning* (i.e. to match their performance). The former directly parameterizes imitation policies as blackbox functions, typically based on behavioral cloning [39–41] or state-action distribution matching [42–44]. The latter takes the indirect approach of first inferring some reward function for which the agent's demonstrations are assumed to be optimal—and on the basis of which an apprentice policy is trained; this is often done via Bayesian [38, 45, 46] or max-ent [47–49] IRL. In contrast, our overarching goal is in *descriptive modeling* (i.e. to learn interpretable parameterizations for explaining observed behavior, see Appendix C for a detailed discussion on our interpretability criteria) [8, 9]. For instance, recent work has represented policies in terms of human-understandable rules [10], goals [11], intentions [12], preferences [13], subjective dynamics [14], and counterfactuals [15].

**Learning from Variation**  Precisely, we wish to understand how behavior has changed over time, especially relevant in healthcare as knowledge [23–25] and practices [16–18] continuously evolve. While *variation in practice* has been captured in policy learning from multi-modal [19, 20], multi-task [21, 22], and compound-task [50, 51] demonstrations, little has sought to describe *variation over time*. Some research has explored IRL from non-stationary agents, using labeled data generated by agents performing specific policy updates over time [36, 37] or ranked post-hoc by an expert [52, 53]. However, all of these simply attempt to infer the optimal reward (viz. ground-truth $\rho_{\mathrm{env}}$) for the (prescriptive) goal of apprenticeship, revealing nothing about *how* the actual behavior has evolved (viz. trajectories $\beta_{1:T}$) for the (descriptive) goal of understanding. The only close attempt has simply used a change-point detection intermediate step to allow inferring a sequence of optimal rewards as usual [54].

**Bandits Setting**  To the best of our knowledge, this is the first formal attempt at learning interpretable representations of non-stationary behavior. In formulating and solving this challenge of learning $\beta_{1:T}$, we have focused on the bandit setting [26–28]. Now, the general notion of an "inverse" bandit problem had been independently proposed by [55] and [56] as the bandit-based counterpart to apprenticeship from a learning agent [36, 37, 52, 53]. However, they both ignore the presence of contexts (viz. "non-contextual" bandits), and—more importantly—identical to the apprenticeship works above, they only attempt to infer the ground-truth reward $\rho_{\mathrm{env}}$, without regard to the trajectory of evolution itself. Table 1 contextualizes ICB with respect to related works in learning from decision-making behavior.

*Table 1. Comparison with Related Work.* We aim to provide an [1] *interpretable* account (i.e. reward-based or white-box) of [2] *non-stationary* behavior (i.e. evolving over time) that explicitly captures the [3] *trajectory* of changes itself (i.e. the agent's knowledge) in a [4] *stepwise* fashion (i.e. versus batched intervals), and [5] *shares information* between consecutive estimates (i.e. $\beta_t, \beta_{t+1}$ are not independent). All works shown operate offline.

| Problem Formulation | | Axis of Variation | Learning Target | Interpretable Output[1] | Non-Stationary Agent[2] | Trajectory of Changes[3] | Stepwise Evolution[4] | Information Sharing[5] | Examples |
|---|---|---|---|---|---|---|---|---|---|
| Policy Learning | Imitation Learning | - | $\pi_b$ | ✗ | ✗ | - | - | - | [40] |
| | Apprenticeship Learning | - | $\rho^*$ | ✓ | ✗ | - | - | - | [57] |
| | Interpretable Policy Learning | - | $\pi_b$ | ✓ | ✗ | - | - | - | [14] |
| Variation in Practice | Multi-Modal Imitation | $K$ clusters | $\{\pi_{b,k}\}$ | ✗ | ✗ | - | - | - | [20] |
| | Compound-Task Learning | $K$ sub-tasks | $\{\pi_k^*\}$ | ✗ | ✗ | - | - | - | [51] |
| | Multi-Task Apprenticeship | $K$ clusters | $\{\rho_k^*\}$ | ✓ | ✗ | - | - | - | [22] |
| Variation over Time | Ranked Reward Extrapolation | $N$ samples | $\rho^*$ | ✓ | ✓ | ✗ | ✗ | ✗ | [52] |
| | Inverse Soft Policy Improvement | $M$ intervals | $\rho^*$ | ✓ | ✓ | ✗ | ✗ | ✗ | [36] |
| | Change-Points Reward Learning | $M$ intervals | $\{\rho_m^*\}$ | ✓ | ✓ | ✓ | ✗ | ✗ | [54] |
| **Inverse Contextual Bandits** | | $T$ time steps | $\rho^*, \{\beta_t\}$ | ✓ | ✓ | ✓ | ✓ | ✓ | **(Ours)** |

# 5 ILLUSTRATIVE EXAMPLES

Three aspects of our approach deserve empirical demonstration, and we shall highlight them in turn:

- *Explainability*: ICB should help us understand how medical practice has changed over the years. This is our primary motivation—in providing interpretable representations of variation over time.
- *Belief Accuracy*: ICB should recover accurate beliefs in a robust manner across a variety of learning agents—that is, it should not be sensitive to the learning procedure of the underlying agent.
- *Reward Accuracy:* ICB should recover accurate ground-truth reward parameters in a similarly robust manner—that is, it should be capable of "extrapolating beyond" suboptimal demonstrations.

**Decision Environments** We consider data from the Organ Procurement & Transplantation Network ("OPTN") as of December 4, 2020, which consists of patients registered for liver transplantation from 1995 to 2020 [58]. We are interested in the decision-making problem of matching organs that become available with patients who are waiting for a transplantation. For each decision, the action space $A_t$ consists of patients who were in the waitlist at the time of an organ's arrival $t$, while the context $x_t[a]$ for each patient $a \in A_t$ includes the features of both the organ and the patient. We consider $k = 8$ features: {ABO mismatch, age, creatinine, dialysis, INR, life support, bilirubin, weight difference}.

In addition to the OPTN dataset, to better validate the performance of our method in a more controlled manner, we also devise a semi-synthetic decision environment, where the features $\{x_t[a]\}_{a \in A_t}$ of potential organ-patient pairs are taken from the OPTN dataset, but the organs are matched with a final patient $a_t \in A_t$ for transplantation according to the policies of a variety of simulated learning agents. We determine the ground-truth reward parameter $\rho_{\text{env}}$ of this semi-synthetic environment by performing simple linear regression over the survival time of patients who actually underwent a transplantation.

**Learning Agents** For the semi-synthetic experiments, we generate observational datasets by employing the following simulated agents, which includes a stationary agent as well as six learning agents:

- *Stationary*: The agent knows the ground-truth $\rho_{\text{env}}$ and takes actions accordingly (i.e. $\rho_t = \rho_{\text{env}}$ for all $t$).
- *Sampling*: The agent employs the posterior sampling-based bandit strategy proposed in [26] (cf. Equation 3).
- *Optimistic*: The agent selects actions based on optimistic estimates of their rewards; this is LinUCB in [27].
- *Greedy*: The agent only exploits their knowledge greedily and does not explore their environment effectively.
- *Stepping*: The agent first explores the environment using uniform preferences until some time step $t^*$, whence they learn the ground-truth reward parameter $\rho_{\text{env}}$ and immediately begin taking actions accordingly. In healthcare settings, stepping behavior like this may occur when new guidelines are introduced.
- *Linear*: The agent learns the true reward parameter $\rho_{\text{env}}$ in a linear fashion, starting with uniform preferences.
- *Regressing*: The agent first acquires the ground-truth reward parameter $\rho_{\text{env}}$ gradually until some time step $t^*$, at which point they begin to regress while retaining some amount of knowledge. Note that this is a particularly challenging setting because the regressing agent's behavior does not improve monotonically.

**Benchmark Algorithms** In addition to our two proposed algorithms Bayesian ICB (**B-ICB**) and Nonparametric Bayesian ICB (**NB-ICB**), we consider all the applicable algorithms in the literature:

- **B-IRL** [38]: This is the classic Bayesian inverse reinforcement learning algorithm that has been widely applied to a variety of problem settings [45, 59, 60]. As usual, it assumes $\rho_t = \rho_{\text{env}}$ for all $t \in \{1, \ldots, T\}$.

- $M$-**fold-IRL**: This runs a copy of Bayesian IRL for each of $M$ equal-sized intervals in the dataset. Note that setting $M = T$ is equivalent to assuming that beliefs over time are completely independent from each other.
- **CP-IRL** [54]: This recently-proposed method runs $M$ copies of Bayesian IRL as well but it employs a change-point detection algorithm to learn best places to divide the dataset (hence referred to as "CP"-IRL).
- **I-SPI** [36]: This recently-proposed method assumes the agent updates their behavioral policy via "soft policy improvements", which can be "inverted" to recover $\rho_{\text{env}}$ (hence referred to as inverse "SPI"). In our setting, this is equivalent to assuming the agent is less stochastic over time (i.e. larger values of $\alpha$). However, I-SPI only recovers the ground-truth $\rho_{\text{env}}$, and does not provide any estimate for belief trajectories.
- **T-REX** [52]: This recently-proposed method learns $\rho_{\text{env}}$ using rankings between context-action pairs; absent explicit rankings, it assumes the agent learns monotonically, such that pairs encountered later in time are preferred over earlier ones (i.e. $(x_t, a_t) \prec (x_{t'}, a_{t'})$ if $t < t'$). Like I-SPI, T-REX only infers $\rho_{\text{env}}$, not beliefs.

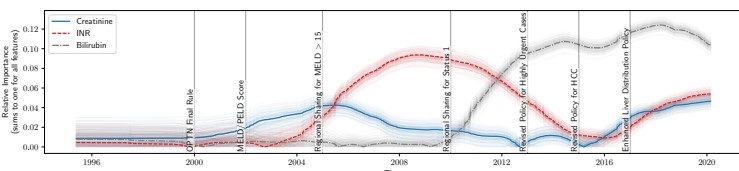

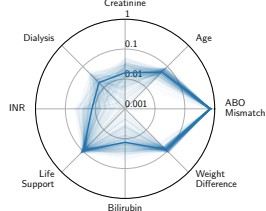

(a) **Feature Importances in 2000**

*Figure 4. Relative Feature Importances over Time for Creatinine, INR, and Bilirubin.* Significant changes in behavior have generally coincided with the important events surrounding guidance on liver allocation policies [61].

In addition to these benchmark algorithms, as a baseline we also report the performance of simply estimating all preferences to be uniform—that is, $\hat{\rho}_{\text{env}} = \hat{\rho}_t = -1/k$ (**Baseline**). Details regarding both the learning agents and the benchmark algorithms can be found in Appendix E.

**Explainability** First, we direct attention to the potential utility of ICB as an *investigative device* for auditing and quantifying behaviors as they evolve. We use NB-ICB to estimate belief parameters $\{\beta_t = \mathbb{E}[\rho_t]\}_{t=1}^T$ for liver transplantations in the OPTN dataset. Since the agent's rewards are linear combinations of features weighted per their belief parameters, we may naturally interpret the normalized belief parameters $|\beta_t(i)|/\sum_{j=1}^k |\beta_t(j)|$ as the *relative importance* of each feature $i \in \{1, \ldots, k\}$. Figure 3 shows the relative importances of all eight features in 2000 and 2010, and Figure 4 shows the importance of creatinine, INR, and bilirubin—components considered in the MELD score ("Model for End-Stage Liver Disease"), a scoring system for assessing

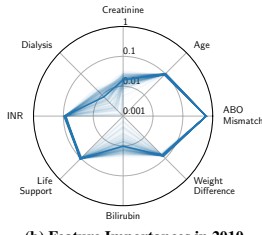

(b) **Feature Importances in 2010**

*Figure 3. Relative Feature Importances in 2000 and 2010.* INR gains significant importance (despite being the least important feature initially), which is consistent with the introduction of MELD in 2002.

the severity of chronic liver disease, [62]. Empirically, three observations immediately stand out: First, INR and creatinine appear to have gained significant importance over the 2000s, despite being the least important features in 2000. Second, their importances appear to have subsequently decreased towards the end of the decade. Third, since 2015 their importances appear to have steadily increased once again.

Interestingly, we can actually verify that these findings are perfectly consistent with the medical environments of their respective time periods. First, the MELD *scoring system* itself was introduced in 2002, which—using INR and creatinine as their most heavily weighted components—explains the rise in importance of those features in the 2000s. Second, over time there was an increase in the usage of MELD *exception points* (i.e. patients getting prioritized for special conditions like hepatocellular carcinoma, which are not directly reflected in their laboratory MELD scores), which explains the decrease in relative importance for such MELD components. Third, 2015 saw the introduction of an *official cap* on the use of MELD exception points (e.g. limited at 34 for hepatocellular carcinoma), which is consistent with the subsequent increase in relative importance of those features once again. Figure 4 also plots important historical events that happened regarding liver allocation policies [61]. Of course, ICB has no knowledge of these events during training, so any apparent changes in behavior in the figure are discovered solely on the basis of organ-patient matching data in the OPTN dataset. Intriguingly, the importance of bilirubin appears to have not increased until 2008, instead of earlier when the MELD score was first introduced. Now, there are possible clinical explanations for this: For instance, bilirubin is not weighted as heavily as other features when computing MELD scores, so their importance may not have been apparent until the later years, when patients generally became much sicker (with higher MELD scores overall). In any case, however, the point here is precisely that ICB is an *investigative device* that allows introspectively describing how policies have changed in

*Table 2. Mean Error of Belief Estimates.* B-ICB and NB-ICB are best across all types of agents. Note that I-SPI and T-REX are not applicable as they do not estimate belief trajectories. Results show mean $\pm$ std. error (5 runs).

| Algorithm | Learning Agent | | | | | | |
|---|---|---|---|---|---|---|---|
| | Stationary | Sampling | Optimistic | Greedy | Stepping | Linear | Regressing |
| Baseline | $0.477 \pm 0.000$ | $0.395 \pm 0.030$ | $0.391 \pm 0.030$ | $0.385 \pm 0.020$ | $0.238 \pm 0.000$ | $0.238 \pm 0.000$ | $0.238 \pm 0.000$ |
| B-IRL | $0.252 \pm 0.248$ | $0.206 \pm 0.099$ | $0.241 \pm 0.195$ | $0.248 \pm 0.204$ | $0.262 \pm 0.027$ | $0.165 \pm 0.019$ | $0.154 \pm 0.018$ |
| 5-fold IRL | $0.294 \pm 0.041$ | $0.369 \pm 0.075$ | $0.313 \pm 0.043$ | $0.310 \pm 0.033$ | $0.333 \pm 0.040$ | $0.289 \pm 0.031$ | $0.292 \pm 0.070$ |
| 10-fold IRL | $0.424 \pm 0.017$ | $0.407 \pm 0.062$ | $0.401 \pm 0.061$ | $0.399 \pm 0.57$ | $0.398 \pm 0.027$ | $0.386 \pm 0.017$ | $0.388 \pm 0.043$ |
| $T$-fold IRL | $1.168 \pm 0.012$ | $1.134 \pm 0.021$ | $1.140 \pm 0.011$ | $1.135 \pm 0.011$ | $1.117 \pm 0.010$ | $1.103 \pm 0.011$ | $1.105 \pm 0.010$ |
| 5-fold CP-IRL | $0.272 \pm 0.040$ | $0.334 \pm 0.039$ | $0.280 \pm 0.029$ | $0.277 \pm 0.029$ | $0.294 \pm 0.046$ | $0.282 \pm 0.029$ | $0.278 \pm 0.010$ |
| 10-fold CP-IRL | $0.409 \pm 0.036$ | $0.462 \pm 0.075$ | $0.388 \pm 0.057$ | $0.384 \pm 0.055$ | $0.374 \pm 0.047$ | $0.376 \pm 0.049$ | $0.401 \pm 0.035$ |
| **B-ICB** | $\mathbf{0.120 \pm 0.032}$ | $\mathbf{0.140 \pm 0.014}$ | $\mathbf{0.121 \pm 0.016}$ | $\mathbf{0.120 \pm 0.018}$ | $0.234 \pm 0.016$ | $0.153 \pm 0.010$ | $0.147 \pm 0.026$ |
| **NB-ICB** | $0.201 \pm 0.027$ | $0.178 \pm 0.017$ | $0.152 \pm 0.042$ | $0.149 \pm 0.036$ | $\mathbf{0.150 \pm 0.012}$ | $\mathbf{0.134 \pm 0.015}$ | $\mathbf{0.140 \pm 0.017}$ |

*Table 3. Mean Error of Ground-Truth Reward Estimates.* B-ICB is best across all types of agents. $M$-fold IRL, CP-IRL, and NB-ICB are not applicable as they do not estimate $\rho_{\text{env}}$. Results show mean $\pm$ std. error (5 runs).

| Algorithm | Learning Agent | | | | | | |
|---|---|---|---|---|---|---|---|
| | Stationary | Sampling | Optimistic | Greedy | Stepping | Linear | Regressing |
| Baseline | $0.477 \pm 0.000$ | $0.477 \pm 0.000$ | $0.477 \pm 0.000$ | $0.477 \pm 0.000$ | $0.477 \pm 0.000$ | $0.477 \pm 0.000$ | $0.477 \pm 0.000$ |
| B-IRL | $0.252 \pm 0.248$ | $0.224 \pm 0.119$ | $0.258 \pm 0.168$ | $0.266 \pm 0.182$ | $0.237 \pm 0.031$ | $0.266 \pm 0.022$ | $0.251 \pm 0.020$ |
| I-SPI | $0.231 \pm 0.212$ | $0.206 \pm 0.092$ | $0.276 \pm 0.208$ | $0.263 \pm 0.183$ | $0.237 \pm 0.031$ | $0.266 \pm 0.021$ | $0.250 \pm 0.020$ |
| T-REX | $1.482 \pm 0.062$ | $1.463 \pm 0.067$ | $1.480 \pm 0.059$ | $1.479 \pm 0.060$ | $1.447 \pm 0.086$ | $1.428 \pm 0.089$ | $1.438 \pm 0.983$ |
| **B-ICB** | $\mathbf{0.121 \pm 0.032}$ | $\mathbf{0.149 \pm 0.050}$ | $\mathbf{0.148 \pm 0.036}$ | $\mathbf{0.141 \pm 0.041}$ | $\mathbf{0.161 \pm 0.027}$ | $\mathbf{0.225 \pm 0.026}$ | $\mathbf{0.246 \pm 0.029}$ |

this manner—such that notable phenomena may be duly investigated with a data-driven starting point (see Appendix C for a discussion on how to interpret behavior with ICB).

**Belief Accuracy** Since it is not possible to compare the belief parameters of different algorithms directly, we define $\|\mathbb{E}_{\rho \sim \mathcal{P}_{\beta_t}}[\rho] - \mathbb{E}_{\rho \sim \mathcal{P}_{\hat{\beta}_t}}[\rho]\|_1$ as the error of belief estimate $\hat{\beta}_t$ at time $t$. Then, Table 2 shows the mean error of belief estimates learned by various algorithms in our semi-synthetic environment. As we would expect, B-ICB performs the best for the Sampling, Optimistic, and Greedy agents—which learn via Bayesian updates—while NB-ICB—which is more flexible in terms how it models belief trajectories—performs the best for the Stepping, Linear, and Regressing agents. Interestingly, we also observe that both $M$-fold IRL and CP-IRL perform worse than vanilla IRL. This could be due to the fact that—in both algorithms—the estimates for each interval are trained with fewer data points and independently from the other intervals; that is, there is no *information sharing* and potential similarities between adjacent intervals are disregarded. In contrast, both B-ICB and NB-ICB formalize some relationship between beliefs at different time steps (viz. Equations 3 and 10); under either formulation, beliefs at different time steps are never independent from each other. Finally, it is worth noting that NB-ICB is capable of capturing the sudden change in the Stepping agent's behavior (despite assuming "smoothly" evolving beliefs, see Appendix A for additional experiments).

**Reward Accuracy** Defining $\|\rho_{\text{env}} - \hat{\rho}_{\text{env}}\|_1$ as the error in estimating the ground-truth parameter $\rho_{\text{env}}$, Table 3 shows the mean error for various algorithms. B-ICB performs the best; this shows that not only can it recover (subjective) descriptors of how the agent *appears to be* behaving over time, but can also infer the (objective) prescriptor of how the agent *ought to be* behaving ideally. That is, B-ICB can also "extrapolate beyond" suboptimal demonstrations to infer the ground-truth reward. Interestingly, T-REX completely fails: This is because in the contextual bandits setting, it effectively assumes later context-action pairs (i.e. generated by a better policy) on average earn larger rewards than earlier ones (i.e. generated by a worse policy)—but this ignores the stochastic arrival of contexts themselves, which (in contextual bandits) is independent of the policy! For instance, sicker patients at a later time may always yield worse outcomes, no matter how much better the allocation policy has become. Appendix A discusses in much more detail why existing methods fail in the ICB setting.

## 6 CONCLUSION

In this paper, we motivated the importance of learning interpretable representations of non-stationary behavior, formalized the problem of ICB, and proposed algorithms with advantages both in explainability and accuracy over any existing methods. Two points deserve brief comment: First, we focused on contextual bandits; while this encapsulates a large class of applications, future work may investigate generalizing this approach to environments with arbitrary $\mathcal{T}$. Second, it is crucial to keep in mind that ICB does not claim to identify the real intentions of an agent: humans are complex, and rationality is bounded. What it does do, is to provide an interpretable explanation of how an agent is effectively behaving, offering a quantitative yardstick by which to investigate hypotheses and understand behavior.

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

# A    DISCUSSION ON EXPERIMENTS

**Applicability of Existing Methods**    Why existing methods should fail or otherwise not apply in our setting requires further exposition. First, note that since ICB is a novel problem (which the proposed B-ICB and NB-ICB algorithms are designed to solve directly), algorithms from prior works will inherently be suboptimal, because they were simply not designed to solve the ICB problem. Concretely, the goal of capturing the *evolution* of non-stationary behavior requires the following (see Table 1):

- **(a) Trajectory of Changes**: Instead of learning a single, static ground-truth reward parameter, we specifically wish to capture the entire trajectory of changes itself.

- **(b) Stepwise Evolution**: If the agent's knowledge evolves continuously over time (i.e. in a step-wise fashion), we wish to capture this (vs. a course, interval-wise appraoch).

- **(c) Information Sharing**: For efficiency in learning, the method should be able to share information between consecutive estimate (i.e. $\beta_t, \beta_{t+1}$ are not independent).

No prior work has been designed with all of these consideration in mind. With respect to Table 2 results (i.e. assessing how well evolving knowledge is learned):

- *B-IRL*: This only learns a single, static ground-truth reward parameter for the entire trajectory, failing all of (a), (b), and (c), hence would underfit.

- *$M$-fold IRL and CP-IRL*: These accomplish (a), but their interval-wise approach fails both (b) and (c), hence would underfit, and is data inefficient.

- *$T$-fold IRL*: This accomplishes (a) and (b), but fails (c) catastrophically by using only one datum to generate each estimate, hence is data inefficient.

With respect to Table 3 (i.e. assessing how well the ground-truth reward is learned), the method should accoutn for non-stationary behavior—that is, instead of assuming the agent is optimal for the learned ground-truth reward the entire time:

- *B-IRL*: This assumes the agent is optimal for the learned ground-truth reward the entire time, which is a model mismatch.

- *I-SPI*: This accounts for non-stationarity, but makes the specific assumption that the behavioral policy is updated via soft policy improvement steps. In the contextual bandits setting, this is equivalent to the assumption that the agent behaves less stochastically as time passes. This is clearly an unreasonably strong assumption, and leads to a model mismatch.

- *T-REX*: This makes the specific assumption that context-action pairs encountered later in time are preferred over earlier ones. In the contextual bandits setting, this is equivalent to the assumption that later context-action pairs (i.e. generated by a better policy) on average earn larger rewards than earlier ones (i.e. generated by a worse policy). But this ignores the stochastic arrival of contexts themselves, which (in contextual bandits) is independent of the policy. For instance, sicker patients at a later time may always yield worse outcomes, no matter how much better the allocation policy has become.

For these reasons, we do not expect existing methods to work, for the simple reason that they were not designed to handle the ICB problem. (And the results corroborate our hypotheses).

**Modeling Sudden Changes**    NB-ICB is capable of identifying sudden changes in behavior since it does not assume a particular mechanism with which the non-stationary behavior has came to be. In this section, we aim to demonstrate this capability with additional experiments. Such sudden changes might occur in clinical settings when new guidelines are introduced or existing guidelines are amended—hence it is critical for our purposes to be able to model them.

Before the introduction of MELD score, the dominant factor of consideration in allocating organs was the waiting time of patients [63]. In fact, MELD was partly introduced to promote the use of features that are more reliable indicators of a patient's liver function (namely creatinine, INR, and bilirubin) than waiting time. Now, consider a hypothetical scenario where a MELD-based policy is introduced at time $t = \lceil T/3 \rceil$ but later withdrawn at time $t = \lfloor 2T/3 \rfloor$ for $T = 2500$. Suppose this causes two sudden changes in clinical practice: (i) Before MELD is introduced, clinicians make decisions based solely on waiting times but they start complying perfectly with the MELD-based

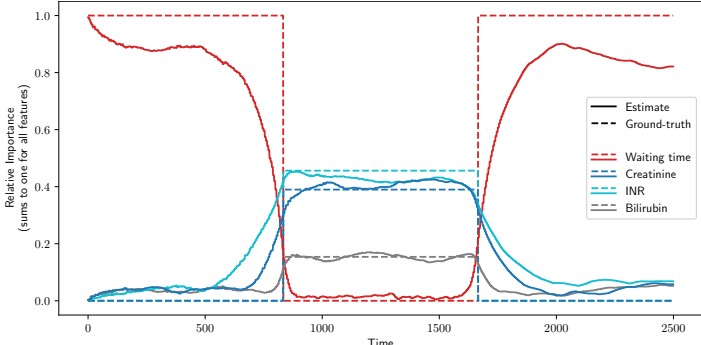

*Figure 5.* Relative feature importances of Waiting time, Creatinine, INR, Bilirubin in the additonal organ allocation scenario as estimated by NB-ICB together with ground-truth feature importances.

| Algorithm | Mean Error |
|---|---|
| B-IRL | $1.158 \pm 0.041$ |
| 5-fold B-IRL | $0.591 \pm 0.025$ |
| 10-fold B-IRL | $0.422 \pm 0.026$ |
| $T$-fold B-IRL | $1.661 \pm 0.006$ |
| 5-fold CP-IRL | $0.310 \pm 0.032$ |
| 10-fold CP-IRL | $0.369 \pm 0.060$ |
| **NB-ICB** | $\mathbf{0.308 \pm 0.024}$ |

*Table 4.* Mean error of belief estimates for the additional organ allocation scenario.

policy once it is introduced. (ii) Later when the MELD-based policy is withdrawn, clinicians revert back to following their original policy based on waiting times again.

In this hypothetical scenario, the context $x[a]$ for a potential organ-patient match consists of four features: {waiting time, creatinine, INR, bilirubin}. Like in our semi-synthetic experiments, we sample these features from the real OPTN data for liver transplantations. For $t \in [T/3, 2T/3]$ (i.e. when the MELD-based policy is in place), $\rho_t$ is such that $\bar{\mathcal{R}}_{\rho_t}(x, a) = 9.57 \log(\text{creatinine}) + 11.2 \log(\text{INR}) + 3.78 \log(\text{bilirubin})$, which matches with the real feature weights used in MELD score [63]. Otherwise, $\rho_t$ is such that $\bar{\mathcal{R}}_{\rho_t}(x, a) = (\text{waiting time})$.

Figure 5 qualitatively shows that NB-ICB is able to identify the sudden changes caused by the introduction and later the withdrawal of the MELD-based policy. Quantitatively, Table 4 show that NB-ICB is still the most accurate algorithm in estimating belief parameters.

## B  DISCUSSION ON NOVELTY AND SETTING

**Problem Setup**  Consider a Markov decision process $\mathbb{D} = (X, A, \mathcal{R}, \mathcal{T})$ as defined in Section 2. In the most general case, the agent has no access to environment dynamics $\mathcal{R}, \mathcal{T}$, so some degree of learning must occur. Let $t$ index the cumulative number of interactions with the environment; if the environment is episodic, then time steps accumulate across resets.

Now, let $f$ denote a probabilistic, online algorithm that learns some parameter $\rho$ of the MDP; for instance, we consider this to be a parameterization of the environment dynamics (but this formalism also applies to learning value functions or black-box policies). And let $\beta_t$ denote the agent's $t$-th step knowledge of the quantity being learned; for instance, this would be a parameterization of their probabilistic belief about the environment dynamics. Note that each $\beta$ naturally induces a policy $\pi_\beta$:

$$\pi_\beta(x)[a] := \mathbb{E}_{\rho \sim \mathcal{P}_\beta}[\pi^*_{\mathcal{R}_\rho, \mathcal{T}_\rho}(x)[a]] , \tag{12}$$

where $\pi^*_{\mathcal{R}_\rho, \mathcal{T}_\rho}$ denotes the optimal policy that corresponds to point-valued knowledge that the environment dynamics are exactly $\mathcal{R}_\rho, \mathcal{T}_\rho$. Finally, let $T$ be the time step when the algorithm $f$ terminates (e.g. due to some measure of convergence). Broadly, $T$ partitions the agent's behavior into two distinct phases:

- *Training phase*: This is where $f$ progressively updates the agent's knowledge by interacting with the environment. Within this phase, we may define the *train-time expected return*:

$$V_{\text{train}}^{\mathcal{R}}(f) := \sum_{t \leq T} \mathbb{E}[r_t \sim \mathcal{R}|\beta_1, \beta_t \sim f, x_t \sim \mathcal{T}, a_t \sim \pi_{\beta_t}] . \qquad (13)$$

- *Testing phase*: Now after training, the agent ends up with final $\beta_T$, and may then be deployed in the environment. Within this phase, we may define the *test-time expected return*:

$$V_{\text{test}}^{\mathcal{R}}(\pi) := \sum_{t > T} \mathbb{E}[r_t \sim \mathcal{R}|x_t \sim \mathcal{T}, a_t \sim \pi] . \qquad (14)$$

Usually, if some knowledge $\beta_T$ were learned in a prior training phase, the agent would naturally set $\pi = \pi_{\beta_T}$ in the testing phase.

**Forward Problem** In the forward direction, an agent seeks to determine the optimal policy that maximizes some notion of expected return. Generally, there are two distinct classes of problems that are of interest: The first assume that agent has access to environment dynamics (either explicitly, via direct knowledge of $\mathcal{R}, \mathcal{T}$; or implicitly, via interaction with the environment in an unassessed training phase). The second assumes the agent has no such access to environment dynamics (therefore learning must occur via interaction with the environment in an assessed training phase).

- **Problem 1A (Agent has dynamics access)**: In this setting, either the dynamics are explicitly known, so no training phase is required (the agent can simply perform planning using dynamic programming, or simulation-based search, etc.), or the agent may freely interact with the environment in an unasses training phase. Either way, $V_{\text{train}}^{\mathcal{R}}$ is irrelevant; the agent simply seeks the following:

$$\pi^* := \operatorname{argmax}_\pi V_{\text{test}}^{\mathcal{R}}(\pi) . \qquad (15)$$

- **Problem 1B (Agent has no dynamics access)**: In this setting, the training phase is assessed as well—that is, not only does the agent seek the optimal policy for test-time deployment, but they also care about the return generated during the training phase. On the flip side, note that once $f$ terminates, $V_{\text{test}}^{\mathcal{R}}$ is trivially minimized for any $T$ by setting $\pi = \pi_{\beta_T}$. Thus the agent seeks the following:

$$f^* := \operatorname{argmax}_f V_{\text{train}}^{\mathcal{R}}(f) . \qquad (16)$$

Crucially, in the former, the agent is optimizing $\pi^*$ based on an "exploitation-only" metric. Whereas in the latter, the agent is required to select an $f^*$ that balances "exploitation" and "exploration".

**Inverse Problem** Now, suppose we—as investigators—are given an observational dataset $\mathcal{D} := \{\boldsymbol{x}_{1:T}, \boldsymbol{a}_{1:T}\}$ of states and actions generated by some agent. The inverse problem deals with determining the true environment parameter $\rho_{\text{env}}$ and/or the agent's belief parameters $\boldsymbol{\beta}_{1:T}$, with respect to which the agent's behavior appears most optimal. This is often treated with a Bayesian formulation:

- **Problem 2A (Agent has dynamics access)**: In this setting, we seek the following:

$$\hat{\rho}_{\text{env}} := \operatorname{argmax}_\rho \mathbb{P}(\mathcal{D}|x_t \sim \mathcal{T}_\rho, a_t \sim \operatorname{argmax}_\pi V_{\text{test}}^{\mathcal{R}_\rho}(\pi))\mathbb{P}(\rho) . \qquad (17)$$

- **Problem 2B (Agent has no dynamics access)**: In this setting, we seek the following:

$$\hat{\rho}_{\text{env}}, \hat{\boldsymbol{\beta}}_{1:T} := \operatorname{argmax}_{\rho, \boldsymbol{\beta}_{1:T}} \mathbb{P}(\mathcal{D}|\beta_1, \beta_t \sim \operatorname{argmax}_f V_{\text{train}}^{\mathcal{R}_\rho}(f), x_t \sim \mathcal{T}_\rho, a_t \sim \pi_{\beta_t})\mathbb{P}(\rho, \boldsymbol{\beta}_{1:T}) . \qquad (18)$$

Note that Problem 2A is equivalent to a generalization of Bayesian inverse reinforcement learning [38]. More broadly, as the "opposite" to Problem 1A, different flavors of inverse reinforcement learning all share the property that they involve—explicitly or implicitly—the quantity $V_{\text{test}}^{\mathcal{R}_\rho}$. For instance, max-margin inverse reinforcement learning [30] seeks $\hat{\rho}_{\text{env}} := \operatorname{argmax}_\rho (V_{\text{test}}^{\mathcal{R}_\rho}(\pi_\mathcal{D}) - \max_\pi V_{\text{test}}^{\mathcal{R}_\rho}(\pi))$.

*On the other hand, Problem 2B is novel, and has not been studied before.* Now, as an initial exploration of this problem setting, in this work we consider state transitions that occur independently of past states and actions (which is especially suited for our motivating application to organ allocations). In the forward sense, this corresponds to a special case of Problem 1B that yields the "contextual bandits"

| | Forward Problem | Inverse Problem |
|---|---|---|
| **Agent has dynamics access** | Problem 1A | Problem 2A |
| (i.e. agent optimizes for exploitation only) | (e.g. any optimal control) | (e.g. Bayesian IRL [38]) |
| **Agent has no dynamics access** | Problem 1B | Problem 2B |
| (i.e. agent balances exploration and exploitation) | (e.g. contextual bandits) | (e.g. Bayesian ICB (Ours)) |

*Table 5.* Classification of problems in sequential decision-making.

setting; this means that decisions can be made greedily: $\mathrm{supp}(\pi^*_{\mathcal{R}_\rho,\mathcal{T}_\rho}(x)) = \mathrm{argmax}_a \bar{\mathcal{R}}_\rho(x,a)$, with ties broken arbitrarily. In the inverse sense, this corresponds to a special case of Problem 2B that yields the "inverse contextual bandits" setting; this means that $\mathcal{T}$ being unknown is ultimately inconsequential, and we may simply treat $\rho$ as the "reward parameter".

Importantly, inverse contextual bandits is *not*—in any shape or form—subsumed by inverse reinforcement learning. Note that B-IRL is an instantiation of Problem 2A, whereas B-ICB is an instantiation of Problem 2B. Table 5 summarizes the above discussion.

## C  FURTHER DISCUSSIONS

**Interpretability**  In terms of interpretability, we are taking the standard view that an interpretable description of behavior should locate the factors that contribute to individual decision, in language readily understood by domain experts [64]. With respect to non-stationary behavior, this concretely manifests in three aspects:

- **(1) Feature Importance**: As is often the case for interpreting supervised learning as well, in representing decision-making a basic desideratum is that the relative importances of inputs be quantified. We argue that linear weights—which we use—are inherently interpretable.

- **(2) Task-level Description**: Specifically for modeling decision-making behavior, however, clearly a task-level description (viz. reward function) is a more concisely interpretable account of the behavior than standard input-output feature sensitivities for a black-box policy model.

- **(3) Non-stationary Behavior**: Finally, the most important factor in our setting is that we specifically wish to describe non-stationary behavior. Note that none of the usual approaches to post-hoc or ante-hoc machine learning interpretability can readily accommodate this unique characteristic.

Given these considerations, let us look at standard approaches to modeling classifiers and/or policies:

- *Input-output feature importance (e.g. [65])*: This satisfies criterion (1), but not (2) or (3).

- *Counterfactual explanations (e.g. [66])*: This satisfies criterion (1), but not (2) or (3).

- *Inverse reinforcement learning (e.g. [67])*: This satisfies criteria (1) and (2), but not (3).

- *Compound-task imitation learning (e.g. [51])*: This satisfies criteria (3), but not (1) or (2).

- *Learning from a learner*: In addition, while I-SPI [36] and T-REX [52] superficially handle non-stationary behavior, they actually only seek to recover the ground-truth reward from such behavior (i.e. $\rho_{\mathrm{env}}$, and do not attempt to describe the evolution of such behavior over time as we do (i.e. $\beta_1, \ldots, \beta_T$)).

In contrast, this is precisely why we propose ICB, which satisfies all three criteria: ICB explicitly seeks to describe how behavior changes over time by learning $\beta_1, \ldots, \beta_T$, which are beliefs over task-level reward functions that give insight into feature importances in the behavior.

It should be emphasized that the primary innovation here is not in proposing any new definition of interpretability. Rather, the novelty is in proposing a formulation and solution to ICB (more generally to Problem 2B in Appendix B) that retains all the advantages of conventional IRL, while—importantly—generalizing to the case where we learn from the non-stationary behavior of an agent with no access to environment dynamics. Of course, we can always naively adapt IRL methods to satisfy criterion (3)—that is, by conducting IRL within discrete intervals within the data: This is simply $M$-fold IRL, and CP-IRL [54], and does not take advantage of the fact that beliefs over time are likely correlated. As corroborated by our experiments, these methods do not appear effective, likely due to the loss of data efficiency from lack of information sharing across time.

**Interpreting Behavior with ICB**   Suppose we were policy-makers, or auditors—in any case, suppose we had a prior *hypothesis* about how behaviors may/should have changed over time. For instance, consider that we introduced a new policy at some time in the past, and would like to verify that it had the intended effect on actual clinical practice. As the OPTN experiment demonstrates, ICB gives a tool for estimating an answer to this question.

In doing so external context (i.e. knoeledge of the times new policies were introduced, or otherwise knowledge of the times at which we hypothesize there to be behavior changes) is crucial. In particular, they should be known *beforehand*. Simply running ICB on a behavioral dataset, and then trying to fish for and explain any trends found, would not be valid use of the method.

Finally, note that ICB does not claim to identify the *real* intentions of an agent: Humans are complex, and rationality is bounded. What it does do, is to provide a representation of how an agent is effectively behaving, offering a quantitative emthod to investigate hypotheses.

**Algorithm 1 vs. Algorithm 2**   Algorithm 1 models the learning procedure of the underlying agent with Bayesian updates and learns both an estimate of the ground-truth reward, as well as the trajectory of beliefs. Whereas, Algorithm 2 models the agent's trajectory of beliefs more generally as a random process but only learns the trajectory of beliefs, without estimating the ground-truth reward. Hence, applying and interpreting the results of each approach depends on the assumptions and specific use case.

For instance, Algorithm 1 makes stronger assumptions, but gives an estimate of $\rho_{\mathrm{env}}$. Since this estimate tells us what the behavioral policy may appear to be optimizing (but is not necessarily perfectly successful at optimizing yet), the estimate can potentially be used to set a new explicit guideline—which, if followed successfullyy—would yield new policies that outperform the existing dataset. Conversely, Algorithm 2 makes weaker assumptions, but the tradeoff is that only a description of the evolutio nof knowledge is recovered, without an estimate of the ground-turth reward for downstream use.

**Forward Agent as an IO-HMM**   The agent in ICB can be viewed as an *Input-Output HMM* (IO-HMM): At any point, the "hidden state" of the model is the agent's belief $\beta_t$, the "input" to the model is the tuple $(x_t, a_t, r_t)$, the "transition function" is the belief-update function $f$, and the "output" of the model is the reward parameter $\rho_t$. In fact, this is depicted in Figure 2 (see the arrows on the top half of the figure).

However, there is a crucial difference: In this IO-HMM, we do not even observe its "outputs" (i.e. $\rho_t$), and even its "inputs" are only partially-observed (i.e. $r_t$). So, learning is much more challenging—be it the IO-HMM parameter $\rho_{\mathrm{env}}$, or the hidden state sequence $\beta_1, \ldots, \beta_T$. Thus even if we chose to cast this problem into a generic IO-HMM framework, it would still be the case that conventional IO-HMM learning techniques are not applicable.

Now, the reason learning is still possible, is that we have additional *prior knowledge* that relates the "inputs" $(x_t, a_t, r_t)$ to the "outputs" $\rho$, such that we can treat them as latent variables for inference. These relationships are also depicted in Figure 2 (see the arrows on the bottom half of the figure). And the algorithms we propose precisely leverage these prior relationships for learning.

**Is ICB itself a bandit algorithm that acts in the same domain?**   The answer is no. There is only ever a single bandit $\mathbb{D} := (X, A, \mathcal{R}, \mathcal{T})$ considered in our setting, which the underlying agent interacts with through a single sequence of actions. There is also only ever a single environment (identified by the tuple $\mathcal{R}, \mathcal{T}$), which the agent starts off having no knowledge about. The agent employs a single belief-update function $f$ to maintain their internal beliefs about the environment at each time step. Each time step, the agent is presented with a new context $x \sim \mathcal{T}$, and takes an action $a \sim \pi_\beta(x)$ based on their policy $\pi_\beta$.

From the perspective of the agent, the forward problem of coming up with a good belief-update function $f$ to maximize rewards in expectation over the entire (single) sequence of interactions is a bandit problem, precisely the "contextual bandit" problem (see Definition 1). From the perspective of the investigator, the inverse problem is simply learning $\rho_{\mathrm{env}}, \boldsymbol{\beta}_{1:T}$ from $\mathcal{D}$, which is the (single) observed sequence of contexts and actions (i.e. "inverse contextual bandits") belongs to an entirely different class problems. Note that, unlike the agent, we as investigators do not interact the environment, observe sequential rewards, nor maintain and update beliefs of any form.

# D    PROOFS OF LEMMAS

**Proof of Lemma 1**    First note that $\mu_t = \Sigma_t(\Sigma_1^{-1}\mu_1 + X_{t-1}\boldsymbol{r}_{1:T})$ by definition. Then, we have

$$\mathbb{P}(\boldsymbol{r}_{1:T}|\boldsymbol{\rho}_{1:T}, \hat{\rho}_{\text{env}}, \hat{\beta}_1, \mathcal{D})$$

$$\propto \mathbb{P}(\boldsymbol{r}_{1:T}, \boldsymbol{\rho}_{1:T}, \hat{\rho}_{\text{env}}, \hat{\beta}_1, \mathcal{D})$$

$$\propto \prod_{t=1}^{T} \mathbb{P}(r_t|\hat{\rho}_{\text{env}}, x_t, a_t) \times \prod_{t=2}^{T} \mathbb{P}(\rho_t|\hat{\beta}_t)$$

$$\propto \prod_{t=1}^{T} \mathcal{R}_{\hat{\rho}_{\text{env}}}(x_t, a_t)[r_t] \times \prod_{t=2}^{T} \mathcal{P}_{\hat{\beta}_t}[\rho_t]$$

$$\propto \mathcal{N}(\sigma^2 X_T^{\mathsf{T}}\hat{\rho}_{\text{env}}, \sigma^2 I)[\boldsymbol{r}_{1:T}] \times \prod_{t=2}^{T} \mathcal{N}(\hat{\mu}_t, \hat{\Sigma}_t)[\rho_t]$$

$$\propto \mathcal{N}(\sigma^2 X_T^{\mathsf{T}}\hat{\rho}_{\text{env}}, \sigma^2 I)[\boldsymbol{r}_{1:T}] \times \prod_{t=2}^{T} \exp\left(-\frac{1}{2} \cdot (\rho_t - \hat{\mu}_t)^{\mathsf{T}} \hat{\Sigma}_t^{-1} (\rho_t - \hat{\mu}_t)\right)$$

$$\propto \mathcal{N}(\sigma^2 X_T^{\mathsf{T}}\hat{\rho}_{\text{env}}, \sigma^2 I)[\boldsymbol{r}_{1:T}]$$
$$\times \prod_{t=2}^{T} \exp\left(-\frac{1}{2} \cdot (\rho_t - \hat{\Sigma}_t\hat{\Sigma}_1^{-1}\hat{\mu}_1 - \hat{\Sigma}_t X_{t-1}\boldsymbol{r}_{1:T})^{\mathsf{T}} \cdot \hat{\Sigma}_t^{-1} \right.$$
$$\left. \cdot (\rho_t - \hat{\Sigma}_t\hat{\Sigma}_1^{-1}\hat{\mu}_1 - \hat{\Sigma}_t X_{t-1}\boldsymbol{r}_{1:T})\right)$$

$$\propto \mathcal{N}(\sigma^2 X_T^{\mathsf{T}}\hat{\rho}_{\text{env}}, \sigma^2 I)[\boldsymbol{r}_{1:T}]$$
$$\times \prod_{t=2}^{T} \exp\left(-\frac{1}{2} \cdot \left(\boldsymbol{r}_{1:T}^{\mathsf{T}} X_{t-1}^{\mathsf{T}} \hat{\Sigma}_t X_{t-1}\boldsymbol{r}_{1:T} - 2(\rho_t - \hat{\Sigma}_t\hat{\Sigma}_1^{-1}\hat{\mu}_1)^{\mathsf{T}} X_{t-1}\boldsymbol{r}_{1:T}\right)\right)$$

$$\propto \mathcal{N}(\sigma^2 X_T^{\mathsf{T}}\hat{\rho}_{\text{env}}, \sigma^2 I)[\boldsymbol{r}_{1:T}]$$
$$\times \prod_{t=2}^{T} \exp\left(-\frac{1}{2} \cdot \left(\boldsymbol{r}_{1:T} - (X_{t-1}^{\mathsf{T}}\hat{\Sigma}_t X_{t-1})^{-1} X_{t-1}^{\mathsf{T}} (\rho_t - \hat{\Sigma}_t\hat{\Sigma}_1^{-1}\hat{\mu}_1)\right)^{\mathsf{T}} \right.$$
$$\left. \cdot X_{t-1}^{\mathsf{T}} \hat{\Sigma}_t X_{t-1} \cdot \left(\boldsymbol{r}_{1:T} - (X_{t-1}^{\mathsf{T}}\hat{\Sigma}_t X_{t-1})^{-1} X_{t-1}^{\mathsf{T}} (\rho_t - \hat{\Sigma}_t\hat{\Sigma}_1^{-1}\hat{\mu}_1)\right)\right)$$

$$\propto \mathcal{N}(\sigma^2 X_T^{\mathsf{T}}\hat{\rho}_{\text{env}}, \sigma^2 I)[\boldsymbol{r}_{1:T}]$$
$$\times \prod_{t=2}^{T} \mathcal{N}\left((X_{t-1}^{\mathsf{T}}\hat{\Sigma}_t X_{t-1})^{-1} X_{t-1}^{\mathsf{T}} (\rho_t - \hat{\Sigma}_t\hat{\Sigma}_1^{-1}\hat{\mu}_1), (X_{t-1}^{\mathsf{T}}\hat{\Sigma}_t X_{t-1})^{-1}\right)[\boldsymbol{r}_{1:T}]$$

$$\propto \mathcal{N}(\sigma^2 X_T^{\mathsf{T}}\hat{\rho}_{\text{env}}, \sigma^2 I)[\boldsymbol{r}_{1:T}]$$
$$\times \mathcal{N}\left(\left(\sum_{t=2}^{T} X_{t-1}^{\mathsf{T}}\hat{\Sigma}_t X_{t-1}\right)^{-1}\left(\sum_{t=2}^{T} X_{t-1}^{\mathsf{T}}(\rho_t - \hat{\Sigma}_t\hat{\Sigma}_1^{-1}\hat{\mu}_1)\right), \right.$$
$$\left. \left(\sum_{t=2}^{T} X_{t-1}^{\mathsf{T}}\hat{\Sigma}_t X_{t-1}\right)^{-1}\right)[\boldsymbol{r}_{1:T}]$$

$$\propto \mathcal{N}\left(\left(\frac{1}{\sigma^2}I + \sum_{t=2}^{T} X_{t-1}^{\mathsf{T}}\hat{\Sigma}_t X_{t-1}\right)^{-1}\left(X_T^{\mathsf{T}}\hat{\rho}_{\text{env}} + \sum_{t=2}^{T} X_{t-1}^{\mathsf{T}}(\rho_t - \hat{\Sigma}_t\hat{\Sigma}_1^{-1}\hat{\mu}_1)\right), \right.$$
$$\left. \left(\frac{1}{\sigma^2}I + \sum_{t=2}^{T} X_{t-1}^{\mathsf{T}}\hat{\Sigma}_t X_{t-1}\right)^{-1}\right)[\boldsymbol{r}_{1:T}].$$

**Proof of Lemma 2**    We have

$$\mathbb{P}(\boldsymbol{\beta}_{1:T}|\boldsymbol{\rho}_{1:T}, \mathcal{D})$$
$$\propto \mathbb{P}(\boldsymbol{\beta}_{1:T}, \boldsymbol{\rho}_{1:T}, \mathcal{D})$$
$$\propto \mathbb{P}(\boldsymbol{\beta}_{1:T}) \cdot \mathbb{P}(\boldsymbol{\rho}_{1:T}|\boldsymbol{\beta}_{1:T}, \mathcal{D})$$

$$\propto \mathcal{N}(\mathbf{0}, \Sigma_T \otimes \Sigma_B)[\text{vec}(\boldsymbol{\beta}_{1:T})] \cdot \mathcal{N}(\text{vec}(\boldsymbol{\beta}_{1:T}), I \otimes \Sigma_P)[\text{vec}(\boldsymbol{\rho}_{1:T})]$$

$$\propto \mathcal{N}(\mathbf{0}, \Sigma_T \otimes \Sigma_B)[\text{vec}(\boldsymbol{\beta}_{1:T})] \cdot \mathcal{N}(\text{vec}(\boldsymbol{\rho}_{1:T}), I \otimes \Sigma_P)[\text{vec}(\boldsymbol{\beta}_{1:T})]$$

$$\propto \mathcal{N}\Big(\big((\Sigma_T \otimes \Sigma_B)^{-1} + (I \otimes \Sigma_P)^{-1}\big)^{-1}(I \otimes \Sigma_P)^{-1}\text{vec}(\boldsymbol{\rho}_{1:T}),$$
$$\big((\Sigma_T \otimes \Sigma_B)^{-1} + (I \otimes \Sigma_P)^{-1}\big)^{-1}\Big)[\text{vec}(\boldsymbol{\beta}_{1:T})] \,.$$

## E  EXPERIMENTAL DETAILS

**Decision Environments**  There are 308,912 patients in the OPTN dataset who either were waiting for a liver transplantation or underwent a liver transplantation. Among these patients, we have filtered out the ones who never underwent a transplantation, the ones who were under the age of 18 or had a donor who was under the age of 18, and the ones who had a missing value for ABO Mismatch, Creatinine, Dialysis, INR, Life Support, Bilirubin, or Weight Difference. This filtering has left us with 31,059 patients, each corresponding to a different donor arrival. For the real-data experiments, we have sampled 2500 donor arrivals uniformly at random, and for each arrival, in addition to the patient who received the donor organ, sampled two more patients who were in the waitlist for a transplantation at the time of donor's arrival in order to form an action space. For the simulated experiments, we have sampled 500 donor arrivals and two patients for each arrival uniformly at random.

**Learning Agents**

- *Sampling:* We set $\sigma = 0.10$.

- *Optimistic:* The agent forms the same posteriors as *Sampling* but selects actions such that $a_t = \text{argmax}_{a \in A} \mu_t^\mathsf{T} x_t[a] + x_t[a]^\mathsf{T} \Sigma_t x_t[a]$.

- *Greedy:* The agent forms the same posterior as *Sampling* but acts based on the mean reward parameters $\rho_t = \mathbb{E}_{\rho \sim \beta_t}[\rho]$.

- *Stepping:* Formally, this agent is given by $\rho_t = -\mathbf{1}/k$ for $t \in \{1, \ldots, t^*\}$ and $\rho_t = \rho_{\text{env}}$ for $t \in \{t^* + 1, \ldots, T\}$. We set $t^* = T/2$.

- *Linear*: Formally, this agent is given by $\rho_t = {}^t/_T \cdot \rho_{\text{env}} + (1 - {}^t/_T) \cdot (-\mathbf{1}/k)$.

- *Regressing*: Formally, this agent is given by $\rho_t = {}^t/_{t^*} \cdot \rho_{\text{env}} + (1 - {}^t/_{t^*}) \cdot \rho^0$ for $t \in \{1, \ldots, t^*\}$ and $\rho_t = {}^{(t-t^*)}/_{(T-t^*)} \cdot \rho^\gamma + (1 - {}^{(t-t^*)}/_{(T-t^*)}) \cdot \rho_{\text{env}}$ for $t \in \{t^* + 1, \ldots, T\}$, where $\rho^0 = -\mathbf{1}/k$ and $\rho^\gamma = \gamma \rho_{\text{env}} + (1 - \gamma)\rho^0$. Here, $\gamma \in [0, 1]$ controls the amount of knowledge that is retained. We set $t^* = T/2$ and $\gamma = 0$.

All learning agents select actions stochastically as described in (5) with $\alpha = 20$.

**Benchmark Algorithms**

- *B-IRL*: We have run the Metropolis-Hastings algorithm for 10,000 iterations to obtain 1,000 samples from $\mathbb{P}(\rho_{\text{env}}|\mathcal{D})$ with intervals of 10 iterations between each sample after 10,000 burn-in iterations. At each iteration, new candidate samples are generated by adding Gaussian noise with covariance matrix $0.005^2 I$ to the last sample. The final estimate $\hat{\rho}_{\text{env}}$ is formed by averaging all samples.

- *M-fold-IRL*: Formally, this assumes $\rho_t = \rho^{(j)}$ for $t \in \{1 + \lfloor (j-1)T/M \rfloor, \ldots, \lfloor jT/M \rfloor\}$ and $j \in \{1, \ldots, M\}$. We have used B-IRL as the IRL solver for each individual fold $j \in \{1, \ldots, M\}$.

- *CP-IRL*: Formally, similar to $M$-fold IRL, this assumes $\rho_t = \rho^{(j)}$ for $t \in (t^{(j-1)}, t^{(j)})$ and $j \in \{1, ...M\}$, where $t^{(0)} = 0$ and $t^{(M)} = T$. However unlike $M$-fold IRL, it employs a change-point detection algorithm to learn the $t^{(j)}$'s together with the $\rho^{(j)}$'s. We have partitioned the trajectory $\{1, \ldots, T\}$ into sub-trajectories of length 10 when detecting change points and used B-IRL as the IRL solver.

- *I-SPI*: We have assumed that the agent performs soft policy improvement after each time step (starting with a uniformly random policy). When computing soft policy improvements, the transition probabilities $\mathcal{T}(x, a)[x'] = \mathcal{T}[x']$ were estimated using the empirical distribution of contexts in dataset $\mathcal{D}$—that is expectations of the form $\mathbb{E}_{x' \sim \mathcal{T}'}[\cdot]$ were approximated by computing $1/T \sum_{x' \in \mathcal{D}}[\cdot]$ instead. Similar to B-IRL, we have run the Metropolis-Hastings algorithm for 10,000

iterations to obtain 1,000 samples from $\mathbb{P}(\rho_{\text{env}}|\mathcal{D})$ with intervals of 10 iterations between each sample after 10,000 burn-in iterations. The same proposal distribution was used, and again, the final estimate $\hat{\rho}_{\text{env}}$ was formed by by averaging all samples.

- *T-REX*: We have maximized the likelihood given in [52] using the Adam optimizer with a learning rate of 0.001, $\beta_1 = 0.9$ and $\beta_2 = 0.999$ until convergence, that is when the likelihood stopped improving for 100 consecutive iterations.

- *B-ICB*: We have set $\sigma = 0.10$, $\alpha = 20$, and $N = 1000$ (with an additional 1000 samples as burn-in). When taking gradient steps, we have used the RMSprop optimizer with a learning rate of 0.001 and a discount factor of 0.9. We have run our algorithm for 100 iterations.

- *NB-ICB*: We have set $\Sigma_P = 5 \cdot 10^{-4} \cdot I$ and $\Sigma_B = 5 \cdot 10^{-5} \cdot I$. We have taken 1,000 samples from $\mathbb{P}(\boldsymbol{\beta}_{1:T}|\mathcal{D})$ with an interval of 10 iterations between each sample after 10,000 burn-in iterations (i.e. $N = 20{,}000$).

For the simulated experiments, the error bars are obtained by repeating each experiment five times.

## SUPPLEMENTARY REFERENCES

[63] R. B. Freeman Jr, R. H. Wiesner, J. P. Roberts, S. McDiarmid, D. M. Dykstra, and R. M. Merion, "Improving liver allcoation: MELD and PELD," *Amer. J. of Transplantation*, vol. 4, pp. 114–131, 2004.

[64] A. Holzinger, C. Biemann, C. S. Pattichis, and D. B. Kell, "What do we need to build explainable ai systems for the medical domain?" *arXiv preprint arXiv:1712.09923*, 2017.

[65] S. Lundberg and S. Lee, "A unified approach to interpreting model predictions," in *Proc. 31st Int. Conf. Neural Inf. Process. Syst.*, 2017.

[66] Y. Goyal, Z. Wu, J. Ernst, D. Batra, D. Parikh, and S. Lee, "Counterfactual visual explanations," in *Proc. 36th Int. Conf. Mach. Learn.*, 2019.

[67] I. Bica, D. Jarrett, A. Hüyük, and M. van der Schaar, "Learning what-if explanations for sequential decision-making," in *Proc. 9th Int. Conf. Learn. Representations*, 2011.

