# OpenReview forum: "Inverse Contextual Bandits: Learning How Behavior Evolves over Time"
_ICLR.cc/2022/Conference — ICLR 2022 Submitted_

### Official Review · Reviewer_pmos · 2021-10-19

**Correctness:** 4
**Technical Novelty And Significance:** 3
**Empirical Novelty And Significance:** 3
**Recommendation:** 6
**Confidence:** 2

**Main Review:**

There are many Strengths in the paper:
- the paper tackles and formulates the important and relevant problem of ICB
- the paper is very well-written and the motivation was easy to follow
- related work is covered rigorously
- simulation setups are clearly stated and seem comprehensive

A weakness might be that the implementations used in the experiments are not shared at this moment. I would love to see an effort to ensure reproducibility (sharing the implementations as supplementary material).

Note here that I’m not the expert in this sub-field, so I might miss some critical merits or weaknesses of the paper.


**Summary Of The Paper:**

The paper formulates and studies the problem named “Inverse Contextual Bandits (ICB)”, which asks “Given demonstrated behavior from a decision-making agent, how has the agent’s knowledge been evolving over time?”  Then, the paper proposes two concrete learning algorithms, imposing different specifications regarding the agent’s behavioral strategy. Finally, some simulations using both simulated and real-world data are provided, showing how ICB can be applied as an investigative device for recovering and explaining the evolution of organ allocation practices over the years.

**Summary Of The Review:**

Overall, the paper is well-written; the motivation, related work, and the experimental part are all easy to follow. I do not find critical flaw in the paper at this moment, but it is very likely that I might miss something. I would recommend weak accept, but I will pay attention to the other reviewer's opinion.

---

> ### Author Response · Authors · 2021-11-19
> **Response to Reviewer pmos**
>
>
> ---
>
> Thank you for your thoughtful comments and suggestions.
>
> We will upload the source code necessary to replicate our experiments as supplementary material as part of the rebuttal revision. Note that running the experiments requires access to the OPTN dataset, which is available to public if requested: [https://optn.transplant.hrsa.gov/data/view-data-reports/request-data/data-request-instructions/](https://optn.transplant.hrsa.gov/data/view-data-reports/request-data/data-request-instructions/).

---

> ### Author Response · Authors · 2021-11-21
> **Dear Reviewer pmos**
>
> Once again, thank you for your invaluable feedback. We were wondering whether our response and the revised manuscript addressed your concerns. If you have any additional comments, please let us know, we would be happy to address them.

---

> > ### Author Response · Authors · 2021-11-27
> > **Dear Reviewer pmos**
> >
> > Thank you very much for your helpful comments. We are sorry to disturb you again but we were just wondering whether our response has adequately addressed your concerns. If not, please let us know what remaining concerns you have; we are very eager to respond to those as well.

---

### Official Review · Reviewer_79eV · 2021-10-28

**Correctness:** 4
**Technical Novelty And Significance:** 2
**Empirical Novelty And Significance:** 2
**Recommendation:** 5
**Confidence:** 4

**Main Review:**

# Abstract

- Style: you don't need to quote the ICB abbreviation; you introduced it for the first time so not only do you not need to quote it that's also not how abbreviations are defined.

# Introduction

- Style: pick quoting or emphasis, don't use both w.r.t. "descriptive modeling".
- I find your core application interesting. I wonder w.r.t. introduced policies if you attach a particular reward to a particular organ allocation policy? Indeed one may find that a policy is not that good at all so then one would have to revert back to the previous observation regime and in so doing, have introduced an unnecessary non-stationary regime.
- Style: please put your desiderata in bullets, it makes it easier to read.
- What do you mean that the environment needs to be learned? I.e. which part are you referring to of the 3-tuple that is a contextual bandit: $\langle A,S,R \rangle$? Please be precise. It is currently not clear what you mean (I think you mean the action-indexed reward-distributions, but pretend that not everyone reading this is a bandit pro).
- Figure 1: I cannot help but to think that this is an overly complex way of depicting non-stationarity. You may as well just show a Markov chain of rewards and simply say that the directed edges propagate through different regimes and ergo the model as stands is non-stationary. Effectively just an HMM wherein each state your agent takes actions. But alas, I may have misunderstood your intentions with that figure.
- Isn't an ICB simply a structural (in time) bandit conditional on a previous bandit which acted in the same domain (though crucially not in the same environment)? Consider bandit one ($t=1$): $\langle A_1,S_1,R_2 \rangle$ and bandit two ($t=2$): $\langle A_1,S_2,R_2 \rangle$ it sounds as if you're suggesting updating the reward distribution at time two as: $p(R_2 \mid A_1, A_2, R_1)$. One bandit is conditional on a past bandit which also took actions in the same system.
- What does "evolves smoothly" mean? Do you mean that you're able to take derivatives and that the environment is Lipschitz continuous (the former is strictly a subset of the latter) or do you mean that there are no discontinuities?

# Inverse contextual bandits

- Notation: perhaps you should stick to convention w.r.t. to the state-space and denote it with $\mathcal{S}$ as is common.
- $\mathbb{D}$ sounds awfully a lot like an MDP - what's the difference?
- "In this work, we consider state transitions that occur independently of past states and actions" - that's an interesting assumption. What's the logic there? We assume that the organ allocation problem operates on the same dynamical system (your 'environment') no? If so, then tracking the non-stationary behaviour, without taking past allocations into account, is akin to saying that your bandits are all independent. But if that is the case why consider contextual bandits at all (I.e. why include $\mathcal{S}$)? Your assumption alone allows you to simply stack up 'normal' bandits $\langle A,R \rangle$ Which it to say you are prescribing more complexity than you are actually using - why?
- It is **not** ok for you to cite research nr [23] (by Y. Qin, F. Imrie etc) which is not yet publically available, not even on arXiv, for us (reviewers) to corroborate your claims.
- You say "Distributions of newly arriving organs are largely independent of prior allocation decisions; [...] [23, 32, 33]" -- figure 2 and section 3 of [33] very much seem to refute your statement. It would be helpful for you to have a similar diagram and point to the d-separation that you claim impose the conditional independence. At present, it sounds implausible.
- For definition 1, one can you please make use of equation environments to show your maths -- you have superscripts to subscripts and it is all becoming a bit too small.
- You have now defined ICB about four times, once is enough.
- Can you please clarify this statement: "In this first work on modeling non-stationary agents," -- non-stationary MABs have been studied for decades so I do not think that's what you mean, hence, please clarify.
- Table 1 is great, good job, very informative.

# Illustrative examples

- Figure 3 is great.
- Figure 4 is great. I think it would be better if you had a similar figure or that figure closer to the top; it really captures the problem you are trying to solve. So placing it closer to the beginning of the paper may help the reader get the gist of your paper early on.

**Summary Of The Paper:**

The authors present the ICB an offline method for providing an interpretable description of observed decision-making whilst capturing the agent's non-stationary understanding of the world.

**Summary Of The Review:**

This is a good and interesting paper. But it is questionable if this could not be achieved with non-stationary MABs or indeed; if past organ allocations have no impact on present allocation, simply running one MAB after another (though it is not clear why we cannot do this). Simply, their novel question in the second paragraph of the introduction is not novel and has been asked many times before.

---

> ### Author Response · Authors · 2021-11-18
> **Response to Reviewer 79eV [Part 9/9]**
>
>
> ---
>
> **(C.6) Style: Equation Environments**
>
> **UPDATE**: We agree; we have now updated the typesetting to use equation environments.
>
> ---
>
> **(C.7) Clarification regarding: "In this first work on modeling non-stationary agents, ..."**
>
> We agree that qualifying this statement would make it more precise.
>
> Here, we are referring to the fact that we are the first to model non-stationary in the **inverse** sense---that is, viz. Problem (P.3.b). **Again, recall that Problem (P.3.b) is novel, and has not been studied before**. (Kindly refer to the Preface to our responses for a detailed contextualization of novelty and setting).
>
> While it is true that MABs have been studied for decades, they have generally only been studied in the **forward** sense---that is, viz. Problem (P.2.b). (Moreover, the only couple of works that have approached any manner of "inverse" problem in the bandit setting have been much simpler [57, 58]: They ignore the presence of contexts, as well as failing to model the evolution of the agent's changes, instead only modeling the ground-truth reward. Kindly refer to our discussion of these and other related works in Section 4).
>
> **UPDATE**: We have updated that phrase, from: "In this first work on modeling non-stationary agents", to: "In this first work on tackling the *inverse* problem in modeling non-stationary agents".
>
> ---
>
> ### (D) Illustrative Examples
>
> ---
>
> **(D.1) Another Figure 4 Closer to the Top**
>
> Thank you for the suggestion. We agree that a similar figure closer to the top would help capture the problem that we are trying to solve.
>
> **UPDATE**: We have now made a simplified version of Figure 4, and used it in place of the original Figure 1, which has now been removed, viz. our Response (B.5).
>
> ---
>
> ### (E) Summary of Review
>
> ---
>
> Thank you for your thoughtful questions and comments.
>
> From your review summary, the main questions are the following:
>
> **(E.1) "It is questionable if this could not be achieved with non-stationary MABs".**
>
> This should now be clear:
>
> - With our Preface clarifying the fundamental difference between **forward** (P.2.b) and **inverse** (P.3.b) problems, as well as our Response (C.7), it should now be obvious that we are tackling the novel (inverse) problem we call ICB, which the well-studied (forward) bandit literature---stationary or otherwise---does not capture at all.
>
> **(E.2) "If past organ allocations have no impact on present allocation", then "it is not clear why we cannot" [...] "simply [run] one MAB after another".**
>
> This should now be clear:
>
> - With our Response (B.6), it should now be clear that the "one MAB after another" perspective is fundamentally incorrect---in particular, because as investigators **we never get to observe** $r_{t}$ (because it is internal to the agent we are attempting to model). To capture the evolution of behavior through beliefs necessarily requires an "inverse" approach.
>
> - Moreover, with our Response (C.3), it should also be clear that it is important to consider **contextual** bandits instead of "normal" multi-armed bandits. In particular, just because the $x_{t}$'s arrive independently, does not mean that we can simply ignore them entirely---the optimal policy for any given context would in general still depend on that context.
>
> **(E.3) "Their novel question in the second paragraph of the introduction is not novel and has been asked many times before".**
>
> This should now be clear:
>
> - The point is the same as with (E.1) above: Again, our Preface now clarifies the fundamental difference between **forward** (P.2.b) and **inverse** (P.3.b) problems. It should now be obvious that we are tackling the inverse problem when we ask "How has the observed behavior changed over time?"---which is novel, and **has not been asked before**. (What has been asked many times before, is simply the forward problem).

---

> ### Author Response · Authors · 2021-11-18
> **Response to Reviewer 79eV [Part 8/9]**
>
>
> ---
>
> **(C.4) Citation is Not Yet Public**
>
> We apologize for this oversight; we had erroneously believed it was already on arXiv at that point.
>
> **UPDATE**: We have now removed it. (In any case, other existing references are already sufficient to corroborate what it also does).
>
> ---
>
> **(C.5) Regarding the correctness of: "Distributions of newly arriving organs are largely independent of prior allocation decisions."**
>
> We politely disagree: This statement is actually **correct**, and the reference in fact supports it.
>
> Kindly allow us to explain. Again, the assumption in question is $X_{t}\perp X_{<t}, A_{<t}$---that is, new organs arrive independently of existing organs and which patients they were matched with. Now:
>
> - First, *on its own this is highly plausible*. Organs arrive according to some external distribution (e.g. occurrences of lethal accidents whose victims supply new organs to the allocation pool). There is no reason whatsoever that this supply of new organs would depend on how prior allocations of organs to patients had been done. (In fact, it is difficult to imagine a realistic scenario in which there is any dependence at all).
>
> - Second, *this formalism is in fact supported by [33]*. See Section 3, paragraph 2 of that paper: "At each time-step, an organ $O_t\in\mathcal{O}$ becomes available for transplant according to some distribution $p(O)$..." [33]. Importantly, this distribution $p(O)$ is unconditional---nowhere in their analysis does it depend on previous allocations. In other words, their setting aligns with ours in that organs arrive independently of prior allocations.
>
> As pertains Figure 2 in [33], there are actually two notational issues that might have given the erroneous impression that the distribution of newly arriving organs depend on other factors---when in fact, they do not. (The following is slightly beyond the scope of our current point of discussion, but for completeness we shall give a brief explanation of what [33] is attempting to do. Note that the following points are the result of a direct correspondence that we had with the authors of [33] to confirm the correctness of our understanding).
>
> - "$X_t$" in the diagram should more precisely read "$X$". That is, the diagram is from the perspective of an *arbitrary* $X$, for which we wish to compute their **potential outcome** $Y$. (Afterwards, using these potential values computed for *all possible such* $X$'s, what [33] does is to propose a forward solution to the matching problem, defining an "optimal policy"---on the basis of these potential values---to select the $X$ that ultimately gets matched to the organ).
>
> - "$O_t$" in the diagram should more precisely read "$O_{X}$". Again, since the diagram is from the perspective of an arbitrary $X$, the organ $O_{X}$ that could be "matched" to $X$ would come from the set $\mathcal{O}\cup\\{\varnothing\\}$ (see Section 3, end of paragraph 2 in [33]). In other words, for each $X$, we can consider two **potential actions**: matching them to the newly arrived organ, or not at all (which would happen if we ultimately match the newly arrived organ to someone else).
>
> Importantly, this organ $O_{X}$ in the middle of the diagram does **NOT** represent a sample from the marginal distribution $p(O)$. Instead, it represents the random variable corresponding to the $X$ under consideration, and can take on the value of either $O_{t}\sim p(O)$ (which, again, is independent of prior allocations), or it can take on the value of $\varnothing$ (which is what would happen if the organ is not allocated to $X$).
>
> **TL;DR**: The statement that "distributions of newly arriving organs are largely independent of prior allocation decisions" is fully consistent with [33].

---

> ### Author Response · Authors · 2021-11-18
> **Response to Reviewer 79eV [Part 7/9]**
>
>
> ---
>
> ### (C) Inverse Contextual Bandits
>
> ---
>
> **(C.1) Notation: $X$ vs. $S$**
>
> Actually, we are simply adhering to the contextual bandits literature in using "$x\in X$" (and referring to them as "contexts"), instead of "$s\in S$" (and referring to them as "states"). While we agree that the latter is more common in the reinforcement learning literature, in the contextual bandits literature the former is in fact standard.
>
> **UPDATE**: We have now added a footnote to the Preliminaries with this notational remark.
>
> ---
>
> **(C.2) Decision-Making Problem**
>
> You are correct: $\mathbb{D}$ is an MDP.
>
> In Section 2, we wished to emphasize the following:
>
> - (a) In typical optimal control, the agent has access to everything: $(X,A,\mathcal{R},\mathcal{T})$.
> - (b) In contextual bandits, the agent only has access to $(X,A)$, and knows nothing about $(\mathcal{R},\mathcal{T})$.
>
> Hence we originally did not want to use the somewhat loaded term "MDP"---because in the literature it often implies setting (a). Whereas, we wanted make sure that it was clear we were operating in setting (b). However, we agree that it is perhaps clearer to simply use the (technically) correct term, which is more precise.
>
> **UPDATE**: We have now changed "decision-making problem" to simply say "Markov decision process".
>
> ---
>
> **(C.3) Independent Transitions**
>
> We politely disagree: We are **not** prescribing more complexity than is needed. In fact, using "normal" bandits would be an incorrect simplification.
>
> Kindly allow us to explain. The assumption in question is $X_{t}\perp X_{<t}, A_{<t}$. Now:
>
> **That is the definition of contextual bandits**. First, the assumption that "state transitions [...] occur independently of past states and actions" is universal to the contextual bandits literature (precisely, that is how they are defined to begin with). There are many use cases for this formalism, and we have not modified or restricted it in any manner at all (again, see e.g. [27]).
>
> **Independence does not imply irrelevance**. Second, we disagree that just because the contexts $x_{t}$ are independent means that we can simply ignore them entirely. Consider the general case of any reward function that depends non-trivially on $x_{t}$ and $a_{t}$. Clearly the optimal action $a_{t}$ should be chosen by taking $x_{t}$ into account, after observing it. So, just because the $x_{t}$'s arrive independently, does not mean that we can simply ignore them entirely (which is what we would do, if we omit $X$ from the formalism and simply treat this as a "normal" bandit problem)...!
>
> What independent contexts implies, is that decisions can be made greedily without consideration of what *future* contexts may be. However, an agent would certainly benefit from taking into consideration what the *current* context is, because it affects the reward.
>
> Finally, specifically as pertains organ allocation, this assumption means that new organs arrive independently of existing organs and which patients they were matched with. To be clear, we are *not* saying that $A_{t}\perp X_{<t}, A_{<t}$, which means something entirely different, and would not be true at all since the decision-maker's internal knowledge evolves over time based on past observations and rewards. So, while the contexts are all independent, **the tuples $(x_t,a_t,r_t)$ are not independent over time**, so a non-contextual bandit formalism would be an incorrect simplification of our setting.

---

> ### Author Response · Authors · 2021-11-18
> **Response to Reviewer 79eV [Part 6/9]**
>
>
> ---
>
> **(B.6) "Isn't an ICB simply a structural (in time) bandit conditional on a previous bandit which acted in the same domain (though crucially not in the same environment)?"**
>
> No, that would be an incorrect characterization. While this is not yet mathematically formalized in the Introduction, in Section 2 we precisely define what is involved in forward/inverse contextual bandits. In particular:
>
> - **Bandit**: There is only ever a single bandit $\mathbb{D}:=(X,A,\mathcal{R},\mathcal{T})$, which the underlying agent interacts with through a single sequence of actions.
> - **Environment**: There is only ever a single environment (identified by the tuple $\mathcal{R},\mathcal{T}$), which the agent starts off having no knowledge about.
> - **Belief-Update**: The agent employs a single belief-update function $f$ to maintain their internal beliefs about the environment at each time step.
> - **Contexts and Actions**: Each time step, the agent is presented with a new context $x\sim\mathcal{T}$, and takes an action $a\sim\pi(x)$ based on their policy $\pi_{\beta}$.
>
> **From the perspective of the agent**: The forward problem (i.e. "contextual bandits"; see Definition 1) is simply coming up with a good $f$ to maximize rewards in expectation over the entire (single) sequence of interactions.
>
> **From the perspective of the investigator**: The inverse problem (i.e. "inverse contextual bandits"; see Definition 2) is simply learning $\rho_{\text{env}},\beta_{1:T}$ from $\mathcal{D}$, which is the (single) observed sequence of contexts and actions.
>
> It is crucial to distinguish between who is making an inference:
>
> - If you are suggesting that the *underlying agent* updates their beliefs about rewards according to $p(\beta_{2}∣a_{1},a_{2},\beta_{1})$, that would be incorrect. Instead, the agent updates their beliefs according to belief-update function $f\in\Delta(B)^{B\times X\times A\times\mathbb{R}}$ (see Definition 1). Note that this formalism for forward contextual bandits problem is completely standard; we have not modified it in anyway (see e.g. [27]).
> - If you are suggesting that we *as investigators* update our beliefs about rewards according to $p(\beta_{2}∣a_{1},a_{2},\beta_{1})$, that would also be incorrect. In order to update our beliefs $\beta_{t}$ about the reward parameter $\rho$ sequentially like that, we must---at a minimum---be able to observe the actual reward values $r_{t}$ to use as input---**but as investigators we never get to observe $r_{t}$, because it is internal to the agent...!**
>
> In contrast, our proposed solution to ICB gets around this, taking into account the entire dataset $\mathcal{D}$ at once to perform Bayesian inference, which detailed in Section 3.
>
> **UPDATE**: We have now included this discussion in the (new) Appendix C, and is now appropriately referenced within Section 2.
>
> ---
>
> **(B.7) "What does "evolves smoothly" mean?"**
>
> As before, while this is not yet formally defined here in the Introduction, in Section 3 we do clarify what we mean by "evolves smoothly". Namely, that:
>
> - "Differences between consecutive beliefs are more or less bounded by $\Sigma_{B}$".
>
> We agree that, to be clearer, this deserves more precise explanation. What we mean is that differences between consecutive beliefs are **probabilistically** bounded by $\Sigma_{B}$:
>
> -  Define $\delta_t:=\beta_{t+1}-\beta_t$. Then we are saying that $\Pr(\delta_t)\propto\exp(-\tfrac{1}{2}\delta_t^{\mathsf{T}}\Sigma_B^{-1}\delta_t)$, independently from other $\delta_t$'s.
>
> In other words, larger changes in consecutive $\beta_{t}$'s are exponentially less likely than smaller changes, in relation to $\Sigma_{B}$.
>
> **UPDATE**: We have now added this comment to the description of "smoothly over time" in Section 3.1, which further clarifies what is meant by this.

---

> ### Author Response · Authors · 2021-11-18
> **Response to Reviewer 79eV [Part 5/9]**
>
>
> ---
>
> **(B.5) Forward Agent as an HMM**
>
> Regarding the agent itself being an HMM-like object, we point out the following nuances:
>
> - You are correct---in the sense that the agent itself can be viewed as an **Input-Output HMM** (IO-HMM): At any point, the "hidden state" of the model is the agent's belief $\beta_{t}$, the "input" to the model is the tuple $(x_{t},a_{t},r_{t})$, the "transition function" is the belief-update function $f$, and the "output" of the model is the reward parameter $\rho_{t}$. In fact, this is depicted exactly in Figure 2 (see the arrows on the top half of the figure).
>
> - However, there is a crucial difference: In this IO-HMM, we do not even observe its "outputs" (i.e. $\rho_{t}$), and even its "inputs" are only partially-observed (i.e. $r_{t}$). So, learning is much more challenging---be it the IO-HMM parameter $\rho_{\text{env}}$, or the hidden state sequence $\beta_{1},...,\beta_{T}$. Thus even if we chose to cast this problem into a generic IO-HMM framework, it would still be the case that **conventional IO-HMM learning techniques are not applicable**.
>
> - Now, the reason learning is still possible, is that we have additional **prior knowledge** that relates the "inputs" $(x_{t},a_{t},r_{t})$ to the "outputs" $\rho$, such that we can treat them as latent variables for inference. These relationships are also depicted in Figure 2 (see the arrows on the bottom half of the figure). And the algorithms we propose precisely leverage these prior relationships for learning.
>
> Regarding Figure 1 being too complex, we agree that it may not be the most intuitive as a pull figure so early in the paper---that is, before adequate notation is introduced. The original intention was simply to highlight the aforementioned distinction that, in our setting, we don't even get to observe the (internal) rewards. However, since Figure 2 contains precisely the same information (but with precise notation), we have now removed Figure 1. (The space for Figure 1 ends up being replaced by another figure that you suggest; see below).
>
> **UPDATE**: We have now included this discussion in a (new) Appendix C, and is now appropriately referenced within Section 2. Furthermore, we have now removed Figure 1, and instead point to Figure 2 to visualize the ICB problem setup.

---

> ### Author Response · Authors · 2021-11-18
> **Response to Reviewer 79eV [Part 4/9]**
>
>
> ---
>
> ### (A) Abstract
>
> ---
>
> **(A.1) Style: Abbreviation**
>
> **UPDATE**: We agree; we have now removed the quotations in the ICB abbreviation.
>
> ---
>
> ### (B) Introduction
>
> ---
>
> **(B.1) Style: Quotation or Italics**
>
> **UPDATE**: We agree; we have now removed the italics in "descriptive modeling".
>
> ---
>
> **(B.2) "[Do we] attach a particular reward to a particular organ allocation policy?"**
>
> You are correct. While this is not yet obvious here in the Introduction, in Section 2 it becomes clear what we mean by capturing "how practices have evolved": Precisely, we formalize this to mean learning the sequence $\beta_{1},...\beta_{T}$ of the agent's beliefs about the reward dynamics (see Definition 2).
>
> In particular, note that each belief $\beta$ induces a policy $\pi_{\beta}$ such that $\pi\_{\beta}(x)[a]:=\mathbb{E}\_{\rho\sim\mathcal{P}\_{\beta}}[\pi\_{\mathcal{R}\_{\rho},\mathcal{T}\_{\rho}}^{*}(x)[a]]$ (see Preface to our responses). So to be absolutely correct, we should say that we are effectively associating a particular *belief about rewards* to a particular organ allocation policy.
>
> You are also correct in that non-stationary behavior can simply be the result of the following: A new policy being introduced, and then being reverted back to the old policy. It is still true that the end result is an agent whose behavior is non-stationary. Note that this is well within the scope of the formalism: Specifically, our non-parametric solution (Algorithm 2) does not assume any specific mechanism behind how non-stationary might have come to be, and it is capable of learning in settings where there are *sudden changes* in policy. To illustrate this fact, we already have experimental results in such a setting (see results for the "Stepping" scenario in Section 5).
>
> Furthermore, in order to highlight this property of Algorithm 2, we have now added new experiments to the appendix (Appendix A). Briefly, we have generated semi-synthetic data like you proposed: (1) Initially, clinicians consider waiting time as the sole factor in making allocation decisions; (2) Then, a MELD-based policy is introduced; (3) However, later the MELD-based policy is withdrawn and clinicians revert back to making decisions based on waiting time. As the new Figure 5 shows, Algorithm 2 is able to find out that a new policy was introduced, as well as the fact that allocation practice then reverted back to their original form sometime later.
>
> **UPDATE**: We have now included this new experiment in **Appendix A**, which is now appropriately referenced within Section 5.
>
> ---
>
> **(B.3) Style: Desiderata in Bullets**
>
> **UPDATE**: We agree; we have now formatted the desiderata using bullets.
>
> ---
>
> **(B.4) "What do [we] mean that the environment needs to be learned?"**
>
> You are correct: In contextual bandits this refers to the reward dynamics $\mathcal{R}\in\Delta(\mathbb{R})^{X\times A}$. While this is already formally denoted in Section 2, we agree that an up-front explanation would benefit the exposition.
>
> **UPDATE**: We have now updated the following phrase (in the Introduction), from: "the environment is unknown to the agent and must be *actively learned*", to: "the environment (i.e. precisely, its reward dynamics) is unknown to the agent and must be *actively learned* by the agent".

---

> ### Author Response · Authors · 2021-11-18
> **Response to Reviewer 79eV [Part 3/9]**
>
>
> ---
>
> **(P.3) Inverse Problem**
>
> Now, suppose we---as investigators---are given an observational dataset $\mathcal{D}:=\{x_{1:T},a_{1:T}\}$ of states and actions generated by some agent. The inverse problem deals with determining the true environment parameter $\rho_{\text{env}}$ and/or the agent's belief parameters $\beta_{1:T}$, with respect to which the agent's behavior appears most optimal. Suppose we treat this with a Bayesian formulation:
>
> - **(a) Agent Has Dynamics Access**: In this setting, we seek the following:
>
> $
> \quad\quad \hat{\rho}\_{\text{env}}:=\text{argmax}\_{\rho}\Pr(\mathcal{D}|x\_{t}\sim\mathcal{T}\_{\rho},a\_{t}\sim\text{argmax}\_{\pi}V_{\text{test}}^{\mathcal{R}\_{\rho}}(\pi))\Pr(\rho)
> $
>
> - **(b) Agent Has No Dynamics Access**: In this setting, we seek the following:
>
> $
> \quad\quad \hat{\rho}\_{\text{env}},\hat{\beta}\_{1:T}:=\text{argmax}\_{\mathcal{\rho},\beta\_{1:T}}\Pr(\mathcal{D}|\beta\_{1},\beta\_{t}\sim\text{argmax}\_{f}V\_{\text{train}}^{\mathcal{R}\_{\rho}}(f),x\_{t}\sim\mathcal{T}\_{\rho},a\_{t}\sim\pi\_{\beta\_{t}})\Pr(\rho,\beta\_{1:T})
> $
>
> Note that Problem (P.3.a) is equivalent to a generalization of Bayesian inverse reinforcement learning [39]. More broadly, as the "opposite" to Problem (P.2.a), different flavors of inverse reinforcement learning all share the property that they involve---explicitly or implicitly---the quantity $V\_{\text{test}}^{\mathcal{R}\_{\rho}}$. For instance, max-margin inverse reinforcement learning [31] seeks $\hat{\rho}\_{\text{env}}:=\text{argmax}\_{\rho}(V\_{\text{test}}^{\mathcal{R}\_{\rho}}(\pi\_{\mathcal{D}})-\text{max}\_{\pi}V\_{\text{test}}^{\mathcal{R}\_{\rho}}(\pi))$.
>
> **On the other hand, Problem (P.3.b) is novel, and has not been studied before**.
>
> Now, as an initial exploration of this problem setting, in this work we consider state transitions that occur independently of past states and actions (which is especially suited for our motivating application to organ allocations).
>
> - In the forward sense, this corresponds to a special case of Problem (P.2.b) that yields the "contextual bandits" setting; this means that decisions can be made greedily: $\text{supp}(\pi\_{\mathcal{R}\_{\rho},\mathcal{T}\_{\rho}}^{*}(x))=\text{argmax}\_{a}\bar{\mathcal{R}}\_{\rho}(x,a)$, with ties broken arbitrarily.
>
> - In the inverse sense, this corresponds to a special case of Problem (P.3.b) that yields the "inverse contextual bandits" setting; this means that $\mathcal{T}$ being unknown is ultimately inconsequential, and we may simply treat $\rho$ as the "reward parameter".
>
> Crucially, in proposing and solving ICB in this work, we are formulating a novel *inverse* problem for non-stationary behavior---viz. Problem (P.3.b). This absolutely **CANNOT** be achieved with non-stationary MABs, which simply formulate the *forward* problem---viz. Problem (P.2.b). Precisely, the question in the second paragraph of the Introduction: "How has the observed behavior changed over time?" **has not been asked before**.
>
> The following table summarizes the above discussion:
>
> ||Forward Problem|Inverse Problem|
> |:--|:--|:--|
> |**Agent Has Dynamics Access** (i.e.       |Problem (P.2.a)         |Problem (P.3.a)         |
> |agent optimizes for exploitation only)    |e.g. any optimal control|e.g. Bayesian IRL [39]  |
> |**Agent Has No Dynamics Access** (i.e.    |Problem (P.2.b)         |Problem (P.3.b)         |
> |agent balances exploration + exploitation)|e.g. contextual bandits |e.g. Bayesian ICB (Ours)|
>
> ---
>
> **UPDATE**: We have now included this discussion in a (new) **Appendix B**, which is now appropriately referenced within Section 2. We have also updated some semantics in the (existing) **Section 2** to minimize possible confusion.

---

> ### Author Response · Authors · 2021-11-18
> **Response to Reviewer 79eV [Part 2/9]**
>
>
> ---
>
> **(P.2) Forward Problem**
>
> In the forward direction, an agent seeks to determine the optimal policy that maximizes some notion of expected return. Generally, there are two distinct classes of problems that are of interest: The first assumes the agent has access to environment dynamics (either explicitly, via direct knowledge of $\mathcal{R},\mathcal{T}$; or implicitly, via interaction with the environment in an unassessed training phase). The second assumes the agent has no such access to environment dynamics (therefore learning must occur via interaction with the environment in an assessed training phase).
>
> - **(a) Agent Has Dynamics Access**: In this setting, either the dynamics are explicitly known, so no training phase is required (the agent can simply perform planning using dynamic programming, or simulation-based search, etc.), or the agent may freely interact with the environment in an unassessed training phase. Either way, $V_{\text{train}}^{\mathcal{R}}$ is irrelevant; the agent simply seeks the following:
>
> $
> \quad\quad \pi^{*}:=\text{argmax}\_{\pi}V\_{\text{test}}^{\mathcal{R}}(\pi)
> $
>
> - **(b) Agent Has No Dynamics Access**: In this setting, the training phase is assessed as well---that is, not only does the agent seek the optimal policy for test-time deployment, but they also care about the returns generated during the training phase. On the flip side, note that once $f$ terminates, $V_{\text{test}}^{\mathcal{R}}$ is trivially minimized for any $T$ by setting $\pi=\pi_{\beta_{T}}$. Thus the agent seeks the following:
>
> $
> \quad\quad f^{*}:=\text{argmax}\_{f}V\_{\text{train}}^{\mathcal{R}}(f)
> $
>
> Note that in the former, the agent is optimizing $\pi^*$ based on an "exploitation-only" metric. Whereas in the latter, the agent is required to select an $f^*$ that balances "exploitation" and "exploration". So far, these concepts are not new: For instance, optimal control corresponds to Problem (P.2.a), and multi-armed bandits are an instance of Problem (P.2.b) where transitions are independent of past states and actions.

---

> ### Author Response · Authors · 2021-11-18
> **Response to Reviewer 79eV [Part 1/9]**
>
>
> ---
>
> Thank you for your thoughtful comments and suggestions. We give answers to each of the following in turn, as well as pointing out corresponding updates to the revised manuscript:
>
> - Preface on Novelty and Setting
> - (A) Abstract
> - (B) Introduction
> - (C) Inverse Contextual Bandits
> - (D) Illustrative Examples
> - (E) Summary of Review
>
> We will upload the revised manuscript shortly after this initial response.
>
> ---
>
> ### Preface on Novelty and Setting
>
> Before we begin, kindly allow us to clarify the context of our setting, which should better elucidate the novelty of our problem and solution.
>
> ICB formulates a novel *inverse* problem for non-stationary behavior---that is, of learning parameterized representations of such behavior from observational data. This absolutely cannot be achieved with non-stationary MABs, which simply formulate the *forward* problem---that is, of determining an optimal strategy given a bandit problem. In fact, the question in the second paragraph of the Introduction: "How has the observed behavior changed over time?" **has not been asked before**.
>
> ---
>
> **(P.1) Problem Setup**
>
> Consider a Markov decision process $(X,A,\mathcal{R},\mathcal{T})$ as defined in the manuscript. In the most general case, the agent has no access to environment dynamics $\mathcal{R},\mathcal{T}$, so some degree of learning must occur. Let $t$ index the cumulative number of interactions with the environment; if the environment is episodic, then time steps accumulate across resets.
>
> Now, let $f$ denote a probabilistic, online algorithm that learns some parameter $\rho$ of the MDP; for instance, we consider this to be a parameterization of the environment dynamics (but this formalism also applies to learning value functions or black-box policies). And let $\beta_{t}$ denote the agent's $t$-th step knowledge of the quantity being learned; for instance, this would be a parameterization of their probabilistic belief about the environment dynamics. Note that each $\beta$ naturally induces a policy $\pi_{\beta}$:
>
> $
> \pi\_{\beta}(x)[a]:=\mathbb{E}\_{\rho\sim\mathcal{P}\_{\beta}}[\pi\_{\mathcal{R}\_{\rho},\mathcal{T}\_{\rho}}^{*}(x)[a]]
> $
>
> where $\pi\_{\mathcal{R}\_{\rho},\mathcal{T}\_{\rho}}^{*}$ denotes the optimal policy that corresponds to point-valued knowledge that the environment dynamics are exactly $\mathcal{R}\_{\rho},\mathcal{T}\_{\rho}$. Finally, let $T$ be the time step when the algorithm $f$ terminates (e.g. due to some measure of convergence). Broadly, $T$ partitions the agent's behavior into two distinct phases:
>
> - **(a) Training Phase**. This is where $f$ progressively updates the agent's knowledge by interacting with the environment. Within this phase, we may define the *train-time expected return*:
>
> $
> \quad\quad V\_{\text{train}}^{\mathcal{R}}(f):=\sum\_{t\leq T}\mathbb{E}[r\_{t}\sim\mathcal{R}|\beta\_{1},\beta\_{t}\sim f,x\_{t}\sim\mathcal{T},a\_{t}\sim\pi\_{\beta\_{t}}]
> $
>
> - **(b) Testing Phase**. Now after training, the agent ends up with a final $\beta_{T}$, and may then be deployed in the environment. Within this phase, we may define the *test-time expected return*:
>
> $
> \quad\quad V\_{\text{test}}^{\mathcal{R}}(\pi):=\sum\_{t>T}\mathbb{E}[r\_{t}\sim\mathcal{R}|x\_{t}\sim\mathcal{T},a\_{t}\sim\pi]
> $
>
> Usually, if some knowledge $\beta_{T}$ were learned in a prior training phase, the agent would naturally set $\pi=\pi_{\beta_{T}}$ in the testing phase.

---

> ### Author Response · Authors · 2021-11-21
> **Dear Reviewer 79eV**
>
> Once again, thank you for your invaluable feedback. We were wondering whether our response and the revised manuscript addressed your concerns. If you have any additional comments, please let us know, we would be happy to address them.

---

> > ### Author Response · Authors · 2021-11-27
> > **Dear Reviewer 79eV**
> >
> > Thank you very much for your helpful comments. We are sorry to disturb you again but we were just wondering whether our response has adequately addressed your concerns. If not, please let us know what remaining concerns you have; we are very eager to respond to those as well.

---

### Official Review · Reviewer_NLig · 2021-11-03

**Correctness:** 3
**Technical Novelty And Significance:** 2
**Empirical Novelty And Significance:** 3
**Recommendation:** 6
**Confidence:** 3

**Main Review:**

Strengths: the considered problem is significant. Their model gains many important properties compared to related works. As they mentioned,  this is the first formal attempt at learning interpretable representations of nonstationary behavior. Using Bayesian methods to compute posteriors is valid. In addition, the experiments for liver transplantations are interesting.

Weaknesses: The presentation is hard to follow. It should be improved.  There is no new technique in this paper and there are no theoretical guarantees for their proposed algorithms. It would be interesting if the authors can upper bound the error $|\rho^* - \hat{\rho}^*|$  to show that their algorithms converge.

**Summary Of The Paper:**

This paper addresses the problem of Inverse Contextual Bandit.  They raise an important question: given demonstrated behavior
from an agent, how has the agent’s knowledge been evolving over time? Formally, Given a contextual bandit problem
$(X, A, R, T )$,  where $R, T$ are unknown to the agent. Given an observational dataset $D$, and a family of reward parameterizations $P$ and belief parameterizations $B$, the inverse contextual bandits problem is to determine the true environment parameter $\rho^*$
and the belief parameters $\beta_{1:T}$.

They propose two algorithms to learn these parameters. The first uses the agent’s knowledge in terms of Bayesian update. The second uses the Gaussian process. They demonstrate their algorithm through simulated and real-world data for liver transplantations.


**Summary Of The Review:**

Based on strengths and weaknesses as I mentioned above, I think this paper is on borderline. Now I am leaning towards acceptance.

---

> ### Author Response · Authors · 2021-11-19
> **Response to Reviewer NLig [Part 2/2]**
>
>
> ---
>
> ### (B) Technical Novelty
>
> ---
>
> First, kindly allow us emphasize that the main novelty of our work is in formulating the ICB problem, which is a particular instance of a brand new class of problems in sequential decision-making (as we discuss in much more detail in the new Appendix B). In particular, this new problem formulation leads to a uniquely structured likelihood function (written in Equation 3), which is not possible to derive using any existing bandit, RL, or IRL formalism. Moreover, even though some existing IRL methods can be (naively) applied to the ICB problem, they are undesirable for various reasons, which we verify empirically (and discuss in more detail in the new Appendix A).
>
> Second, regarding theoretical guarantees, note that we optimize Equation 3 (Equation 5 in the revised manuscript) using the standard expectation-maximization (EM) algorithm. Now, if computing the *exact* expected log-likelihood (i.e. $\mathcal{Q}$) were to be possible analytically, then via standard results this approach would have been guaranteed to converge (to some local maxima) as our likelihood function is continuous and differentiable. However, in our case the particular $\mathcal{Q}$ we need to compute is intractable, hence we resort to approximating it via sampling. That said, note that it follows from Glivenko–Cantelli theorem that as we increase the number of samples $N$, the maximum difference between $\mathcal{Q}$ and the approximated $\mathcal{Q}$ (i.e.$\bar{\mathcal{Q}}$) tends to zero [A]. Moreover, it is also worth noting that approximating $\mathcal{Q}$ via sampling is an established approach in the literature [B,C].
>
> ---
>
> **References**
>
> [A] R. Durrett, *Probability: theory and examples*, Cambridge University Press, 2010.
>
> [B] R. A. Levine and G. Casella, “Implementations of the Monte Carlo EM algorithm,” J. Comput. Graph. Statist., vol. 10, no. 3, pp. 422-439, 2001.
>
> [C] R. C. Neath, “On convergence properties of the Monte Carlo EM algorithm,” 10.1214/12-IMSCOLL1003, 2012.

---

> ### Author Response · Authors · 2021-11-19
> **Response to Reviewer NLig [Part 1/2]**
>
>
> ---
>
> Thank you for your thoughtful comments and suggestions. We give answers to each in turn, as well as pointing out corresponding updates to the revised manuscript:
>
> - (A) Presentation
> - (B) Technical Novelty
>
> We will upload the revised manuscript shortly after this initial response.
>
> ---
>
> ### (A) Presentation
>
> ---
>
> We agree that our presentation can be dense at times. This is because ICB combines many different strands of research (bandits, interpretability, variation in practice, and inverse reinforcement learning), while at the same time proposing and solving a novel problem, so the limited space has required being very concise.
>
> To enhance readability by providing easy reference for additional background as to how ICB relates to these topics, we have now added **three new appendices** discussing the following topics in detail:
>
> - **Appendix A: Subsection “Discussion on Applicability of Existing Methods”**: This describes why exactly the existing benchmarks fail or do not perform as well as B-ICB/NB-ICB does, in our problem setting. The short answer is: None of the existing methods are designed specifically for the ICB setting. In Appendix A, we go into much more detail, explaining what aspects of ICB each existing method overlooks one by one and how it leads to their suboptimal performance.
>
> - **Appendix B: Discussion on Novelty and Setting**: This clarifies how conventional formulations of reinforcement learning (RL), inverse reinforcement learning (IRL), and contextual bandits (CB) relate to our novel formalism that is the inverse contextual bandits problem. There, we define mathematically what “having unrestricted access to dynamics” means, and precisely compare and contrast problems in sequential decision-making into four classes: Forward problems with access to dynamics (Problem 1A), forward problems with no access to dynamics (Problem 1B), inverse problems with access to dynamics (Problem 2A), and inverse problems with no access to dynamics (Problem 2B). Then, RL is a particular instance of Problem 1A, IRL of Problem 2A, and CB of Problem 1B. Importantly, this precisely identifies ICB as the first instance of Problem 2B ever considered.
>
> - **Appendix C: Subsection “Discussion on Interpretability”**: This defines our concrete criteria for interpretability that we desire of any solution to ICB. It also discusses how to interpret the results we present in Section 5, as well as how Algorithm 1 and 2 (B-ICB and NB-ICB) provide different interpretations that could be beneficial for different use cases.
>
> Finally, we now reference these new appendices in the main text where appropriate, which should enhance readability and background reference. We have also updated some semantics in the existing Section 2 to minimize possible confusion regarding the relationship between RL, IRL, CB, and ICB.

---

> ### Author Response · Authors · 2021-11-21
> **Dear Reviewer NLig**
>
> Once again, thank you for your invaluable feedback. We were wondering whether our response and the revised manuscript addressed your concerns. If you have any additional comments, please let us know, we would be happy to address them.

---

> > ### Author Response · Authors · 2021-11-27
> > **Dear Reviewer NLig**
> >
> > Thank you very much for your helpful comments. We are sorry to disturb you again but we were just wondering whether our response has adequately addressed your concerns. If not, please let us know what remaining concerns you have; we are very eager to respond to those as well.

---

### Official Review · Reviewer_G5nF · 2021-11-08

**Correctness:** 3
**Technical Novelty And Significance:** 2
**Empirical Novelty And Significance:** 2
**Recommendation:** 5
**Confidence:** 4

**Main Review:**

**Problem definition and setup**:

I found much of the content in section 2 to be unnecessarily general. Indeed, the preliminaries describe a reinforcement learning framework, but only the contextual bandits setting is considered in the paper. I found this additional notation to be distracting to the paper, and would suggest the authors introduce only background that is used in the paper. Additionally, this unneeded generality makes explanations in various parts of the paper more complicated than needed -- in particular, Remark 1 would not be needed if the RL framework were not introduced.

Additionally, I found the objectives of the paper to be rather vague. Specifically, the authors place significant emphasis on the interpretability of the output of an ICB algorithm, but do not seem to describe what interpretability means. I think it would be useful to describe some desired or defining characteristics of an interpretable output, and to discuss how the belief trajectory, as the authors have modelled it in their two algorithms, does or does not satisfy those characteristics.

**Relation to related work**:

I am a bit confused with how this work fits into/differs from some prior works mentioned in the paper. In particular, I was quite confused why one of the three main contributions of the paper was to formalize the ICB problem, but the IRL problem (which, as I understand it, is strictly more general) has already been formalized and studied. Could you elaborate more on how your problem setting and approaches differ from, e.g., [39] (Bayesian inverse reinforcement learning)? In what ways are you solving a different problem, or improving on previous works on IRL in your contextual bandits setting? Why should one expect the performance in Tables 3-4 of the algorithms you propose to be better than those from prior works (e.g., B-IRL)?

**Experimental results**:

While I find it interesting that, in the OPTN experiment, you are able to associate (external) policy decisions with changes in feature importances, I have a number of questions/comments about the interpretability of the algorithm output, and the applicability of the model considered in the paper.

Contextual bandits model: I think it would be useful to explain (i) what the rewards represent in the context of this experiment (I couldn’t find this explained anywhere, but I might have missed it) and (ii) why it is reasonable to assume that $\rho^*$ does not vary over time. I was confused on point (ii) for a while, since I would expect most bandits algorithms to converge to a fixed $\rho^*$. In the context of your experiments, this would correspond to the relative importance of the features converging to a single value. Could you explain why one should not expect this to happen in your experiments?

Explainability of the results: I think that the paper could benefit from more explicitly outlining the desired properties of explainable results, how their method achieves this, and various ways that the method could be used. Indeed, in order for the results of Figures 3-4 to be “explainable” or “interpretable,” it seems that external context (such as that given in Figure 4) is crucial. Even then, only general trends can be seen from the figures displayed in this section. It is not clear to me how one might interpret such trends in this way when the number of features is large. Do the authors have more general ideas for interpreting these results? How might interpreting these results change when using Algorithm 1 vs Algorithm 2 in your paper? It may be useful to discuss these in the paper.


**Summary Of The Paper:**

This paper studies an inverse (linear) contextual bandits (ICB) problem, where, given a $T$-round realization of a bandit policy’s actions and observed rewards, the goal is to design an algorithm to estimate the underlying environment parameter, along with the “belief trajectory” of the bandit policy. A particular emphasis is placed on the belief trajectory being “interpretable” and capturing changes in the policy’s “knowledge of the world” over time.

The paper’s main contributions are (i) formalizing the inverse contextual bandits problem, (ii) designing two algorithms for this problem based on two different ways of modelling beliefs of the bandit policy, and (iii) providing empirical illustrations of how their algorithm can be used to investigate and explain changes in medical decision-making over time


**Summary Of The Review:**

While the problem setting seems interesting, I think that there are a number of issues indicating that this work is not yet ready for publication. Indeed, the problem setting should be more concisely described, and the relation of this work to prior works on IRL should be discussed more thoroughly. Additionally, I think that the authors should more clearly outline the goals of the interpretability of their method, and outline concrete solutions others could use to interpret the belief trajectories in practice.

---

> ### Author Response · Authors · 2021-11-17
> **Response to Reviewer G5nF [Part 9/9]**
>
>
> ---
>
> ### (E) Algorithm 1 vs. Algorithm 2
>
> ---
>
> Thank you for the question.
>
> Algorithm 1:
>
> - models the learning procedure of the underlying agent with Bayesian updates
> - learns both an estimate of the ground-truth reward, as well as the trajectory of beliefs
>
> Algorithm 2:
>
> - models the agent's trajectory of beliefs more generally as a random process
> - only learns the trajectory of beliefs, without estimating the ground-truth reward
>
> Hence, applying and interpreting the results of each approach depends on the assumptions and specific use case. For instance, Algorithm 1 makes stronger assumptions, but gives an estimate of $\rho_{\text{env}}$. Since this estimate tells us what the behavioral policy may appear to be optimizing (but is not necessarily perfectly successful at optimizing yet), the estimate can potentially be used to set a new explicit guideline---which, if followed successfully---would yield new policies that outperform the existing dataset. Conversely, Algorithm 2 makes weaker assumptions, but the tradeoff is that only a description of the evolution of knowledge is recovered, without an estimate of the ground-truth reward for downstream use.
>
> **UPDATE**: We have now included this discussion in the (new) **Appendix C**, and is now appropriately referenced within Section 3.

---

> ### Author Response · Authors · 2021-11-17
> **Response to Reviewer G5nF [Part 8/9]**
>
>
> ---
>
> ### (D) Interpreting the Experiments
>
> ---
>
> **(D.1) What do the rewards represent in the context of the experiment?**
>
> Thank you for the question.
>
> We agree that explicitly stating this would clarify the interpretation of the experiment. A reward function in the OPTN experiment represents a subjective measure of how "desirable" a given match is, from the perspective of the evolving behavioral policy. Therefore, as investigators seeking to understand the behavioral policy, learning about a policy's reward function measures tells us how "beneficial" an organ for a patient likely appeared to the decision-maker. Moreover, since we operate in the space of linear reward functions, the weights capture the importance of each feature in determining the desirability of a match.
>
> **UPDATE**: We have now included this clarification in Section 5.
>
> ---
>
> **(D.2) Why is it reasonable to assume $\rho_{\text{env}}$ does not vary over time?**
>
> Thank you for the question.
>
> You are correct that if $\rho_{\text{env}}$ were static (as it is often the case in bandit problems), standard bandit algorithms would converge to that. For instance, Thompson sampling would converge to $\rho_{\text{env}}$.
>
> - If we knew that this was indeed the case, then Algorithm 1 (Bayesian ICB) would be appropriate, since it models exactly for this scenario.
>
> However, you are also correct that it may not always be reasonable to assume $\rho_{\text{env}}$ is static---or that there was indeed any $\rho_{\text{env}}$ that induced the behavioral policy to begin with. Indeed, in OPTN this would constitute a strong assumption, which is why we do not make such an assumption for the OPTN experiment.
>
> - Since we are not willing to make this assumption, we have used Algorithm 2 for the OPTN experiment, which makes the milder assumption that the agent’s behavior evolves smoothly over time (and dispenses with learning a single, static ground-truth reward). Note that this no longer assumes there is any $\rho_{\text{env}}$ to which the underlying agent is learning to converge towards.
>
> ---
>
> **(D.3) Explainability of the results**
>
> Thank you for the question.
>
> Regarding what is desirable for interpretability, please kindly refer to our Response (B): "Criteria for Interpretability", where have now explicitly identified the desiderata for an ideal method, discuss how existing works do not accomplish them, as well as how ICB does satisfy them.
>
> Regarding how this method should be used, you are absolutely correct that external context is crucial for applicability of this method. Allow us to clarify what we mean by ICB being an "investigative device for auditing and quantifying behaviors as they evolve":
>
> - Suppose we were policy-makers, or auditors---in any case, suppose we had a prior *hypothesis* about how behaviors may/should have changed over time. For instance, consider that we introduced a new policy at some time in the past, and would like to verify that it had the intended effect on actual clinical practice. As the OPTN experiment demonstrates, ICB gives a tool for estimating an answer to this question.
> - You are correct that the external context (i.e. knowledge of the times new policies were introduced, or otherwise knowledge of the times at which we hypothesize there to be behavioral changes) is crucial. In particular, they should be known *beforehand*. Simply running ICB on a behavioral dataset, and then trying to fish for and explain any trends found, would not be a valid use of the method.
> - Finally, note (as we do in the Conclusion) that ICB does not claim to identify the *real* intentions of an agent: Humans are complex, and rationality is bounded. What it does do, is to provide a representation of how an agent is effectively behaving, offering a quantitative method to investigate hypotheses.
>
> **UPDATE**: We have now included this discussion in the (new) **Appendix C**, and is now appropriately referenced within Section 5.

---

> ### Author Response · Authors · 2021-11-17
> **Response to Reviewer G5nF [Part 7/9]**
>
>
> ---
>
> **(C.2) Why should ICB perform better than prior works?**
>
> Thank you for the question.
>
> We agree that an explicit discussion, of why existing methods should fail or otherwise not apply, would benefit the exposition.
>
> First, note that since ICB is a novel problem (which the proposed B-ICB and NB-ICB algorithms are designed to solve directly), algorithms from prior works will inherently be suboptimal, because they were simply not designed to solve the ICB problem. Concretely, the goal of capturing the *evolution* of non-stationary behavior requires the following (see Table 1):
>
> - **(a) Trajectory of Changes**. Instead of learning a single, static ground-truth reward parameter, we specifically wish to capture the entire trajectory of changes itself.
> - **(b) Stepwise Evolution**. If the agent's knowledge evolves continuously over time (i.e. in a step-wise fashion), we wish to capture this (vs. a course, interval-wise approach).
> - **(c) Information Sharing**. For efficiency in learning, the method should be able to share information between consecutive estimates (i.e. $\beta_{t},\beta_{t+1}$ are not independent).
>
> No prior work has been designed with all of these considerations in mind. With respect to Table 2 results (i.e. assessing how well evolving knowledge is learned):
>
> - **B-IRL**: This only learns a single, static ground-truth reward parameter for the entire trajectory, failing all of (a), (b), and (c), hence would underfit.
> - **M-fold IRL** and **CP-IRL**: These accomplish (a), but their interval-wise approach fails both (b) and (c), hence would underfit, and is data inefficient.
> - **T-fold IRL**: This accomplishes (a) and (b), but fails (c) catastrophically by using only one datum to generate each estimate, hence is data inefficient.
>
> With respect to Table 3 results (i.e. assessing how well the ground-truth reward is learned), the method should account for non-stationary behavior---that is, instead of assuming the agent is optimal for the learned ground-truth reward the entire time:
>
> - **B-IRL**: This assumes the agent is optimal for the learned ground-truth reward the entire time, which is a model mismatch.
> - **I-SPI**: This accounts for non-stationarity, but makes the specific assumption that the behavioral policy is updated via soft policy improvement steps. In the contextual bandits setting, this is equivalent to the assumption that the agent behaves less stochastically as time passes. This is clearly an unreasonably strong assumption, and leads to a model mismatch.
> - **T-REX**: This makes the specific assumption that context-action pairs encountered later in time are preferred over earlier ones. In the contextual bandits setting, this is equivalent to the assumption that later context-action pairs (i.e. generated by a better policy) on average earn larger rewards than earlier ones (i.e. generated by a worse policy). But this ignores the stochastic arrival of contexts themselves, which (in contextual bandits) is independent of the policy! For instance, sicker patients at a later time may always yield worse outcomes, no matter how much better the allocation policy has become.
>
> For these reasons, we do not expect existing methods to work, for the simple reason that they were not designed to handle the ICB problem. (And the results corroborate our hypotheses). While some of these points are already made in the manuscript, we agree that a targeted discussion is helpful to understand the experiments.
>
> **UPDATE**: We have now included this discussion in **Appendix A**, which is now appropriately referenced within Section 5.

---

> ### Author Response · Authors · 2021-11-17
> **Response to Reviewer G5nF [Part 6/9]**
>
>
> ---
>
> ### (C) Relation to Related Work
>
> ---
>
> **(C.1) How does IRL relate to ICB?**
>
> Thank you for the question.
>
> From the discussion in the above Preface to our responses, the precise context of ICB and our contributions should now be clear. In particular, it should now be clear how IRL relates to ICB. The following remarks reproduce parts of our Preface and Response (A) above:
>
> > The RL and IRL frameworks are actually *not sufficiently general* for our purposes---because they do not accommodate non-stationarity of underlying agents who have no dynamics access. On the one hand:
> >
> > - RL as described: This corresponds to **Problem (P.2.a)** and variants thereof.
> > - IRL as described: This corresponds to **Problem (P.3.a)** and variants thereof.
> >
> > On the other hand, we seek to learn from data how behavior has evolved over time:
> >
> > - In the forward sense our setting actually requires **Problem (P.2.b)**, where an agent approaches a decision-making problem without unrestricted access to the environment's dynamics---which is not captured by the RL formalism.
> > - In particular, in the inverse sense we are interested in the novel **Problem (P.3.b)**, where we (as investigators) seek to learn from the non-stationary behavior of such an underlying agent---which is not captured by the IRL formalism.
> >
> > Importantly, we reiterate the following:
> >
> > - ICB---which corresponds to Problem (P.3.b)---is **NOT** subsumed by IRL---which corresponds to Problem (P.3.a).
> > - Bayesian ICB---which instantiates Problem (P.3.b)---is **NOT** subsumed by Bayesian IRL---which instantiates Problem Problem (P.3.a).
> >
> > (Of course, one can indeed imagine studying Problem (P.3.b) more generally in the case of arbitrary transition dynamics; but as noted in the manuscript, we leave that to future work).

---

> ### Author Response · Authors · 2021-11-17
> **Response to Reviewer G5nF [Part 5/9]**
>
>
> ---
>
> ### (B) Criteria for Interpretability
>
> ---
>
> Thank you for the suggestion.
>
> We agree that explicitly identifying the criteria for an ideal method would benefit the exposition. Before we begin, allow us to emphasize that the primary innovation here is not in proposing any new definition of interpretability. Rather, the novelty is in proposing a formulation and solution to Problem (P.3.b) that retains all the advantages of conventional IRL, while---importantly---generalizing to the case where we learn from the non-stationary behavior of an agent with no access to environment dynamics.
>
> In terms of interpretability, we are taking the standard view that an interpretable description of behavior should locate the factors that contribute to individual decisions, in a language readily understood by domain experts [64]. With respect to non-stationary behavior, this concretely manifests in three aspects:
>
> - **(1) Feature Importance**. As is often the case for interpreting supervised learning as well, in representing decision-making a basic desideratum is that the relative importances of inputs be quantified. We argue that linear weights---which we use---are inherently interpretable.
> - **(2) Task-level Description**. Specifically for modeling decision-making behavior, however, clearly a task-level description (viz. reward function) is a more concisely interpretable account of the behavior than standard input-output feature sensitivities for a black-box policy model.
> - **(3) Non-stationary Behavior**. Finally, the most important factor in our setting is that we specifically wish to describe non-stationary behavior. Note that none of the usual approaches to post-hoc or ante-hoc machine learning interpretability can readily accommodate this unique characteristic.
>
> Given these considerations, let us look at standard approaches to modeling classifiers and/or policies:
>
> - **Input-Output Feature Importance** (e.g. [65]): This satisfies criterion (1), but not (2) or (3).
> - **Counterfactual Explanations** (e.g. [66]): This satisfies criterion (1), but not (2) or (3).
> - **Inverse Reinforcement Learning** (e.g. [67]): This satisfies criteria (1) and (2), but not (3).
> - **Compound-Task Imitation Learning** (e.g. [52]): This satisfies criteria (3), but not (1) or (2).
> - **Learning from a Learner**: In addition, while I-SPI [37] and T-REX [53] superficially handle non-stationary behavior, they actually only seek to recover the ground-truth reward from such behavior (i.e. $\rho_{\text{env}}$), and do not attempt to describe the evolution of such behavior over time as we do (i.e. $\beta_{1},...,\beta_{T}$).
>
> In contrast, this is precisely why we propose ICB, which satisfies all three criteria: ICB explicitly seeks to describe how behavior changes over time by learning $\beta_{1},...,\beta_{T}$, which are beliefs over task-level reward functions that give insight into feature importances in the behavior.
>
> (Of course, we can always naively adapt IRL methods to satisfy criterion (3)---that is, by conducting IRL within discrete intervals within the data: This is simply M-fold IRL, and CP-IRL [55], and does not take advantage of the fact that beliefs over time are likely correlated. As corroborated by our experiments, these methods do not appear effective, likely due to the loss of data efficiency from lack of information sharing across time).
>
> **UPDATE**: We have now included this discussion in a (new) **Appendix C**, which is now appropriately referenced within Section 1 and 4.
>
> ---
>
> **References**:
>
> - [64] A. Holzinger, C. Biemann, C. Pattichis, and D. Constantinos, "What Do We Need to Build
> Explainable AI Systems for the Medical Domain?", 2017.
> - [65] S. Lundberg, S. Lee, "A Unified Approach to Interpreting Model Predictions", 2017.
> - [66] Y. Goyal, Z. Wu, J. Ernst, D. Batra, D. Parikh, S. Lee, "Counterfactual Visual Explanations", 2019.
> - [67] I. Bica, D. Jarrett, A. Hüyük, and M. van der Schaar, "Learning What-If Explanations for Sequential Decision-Making", 2021.

---

> ### Author Response · Authors · 2021-11-17
> **Response to Reviewer G5nF [Part 4/9]**
>
>
> ---
>
> ### (A) Generality of Preliminaries
>
> ---
>
> Thank you for the comment.
>
> We agree that the terseness of the original exposition may have left the erroneous impression that the conventional RL/IRL frameworks described in the Preliminaries are unnecessarily general for our purposes. However, the opposite is actually true: From the preceding discussion in the Preface to our responses, it should now be clear that these frameworks are in fact *not sufficiently general* for our purposes---because they do not accommodate non-stationarity of underlying agents who have no dynamics access:
>
> - RL as described: This corresponds to **Problem (P.2.a)** and variants thereof.
> - IRL as described: This corresponds to **Problem (P.3.a)** and variants thereof.
>
> However:
>
> - In the forward sense our setting actually requires **Problem (P.2.b)**, where an agent approaches a decision-making problem without unrestricted access to the environment's dynamics---which is not captured by the RL formalism.
> - In particular, in the inverse sense we are interested in the novel **Problem (P.3.b)**, where we (as investigators) seek to learn from the non-stationary behavior of such an underlying agent---which is not captured by the IRL formalism.
>
> So the goal of Section 2 is the following:
>
> - (1) Set up the RL and IRL frameworks---which, again, correspond to Problems (P.2.a) and (P.3.a).
> - (2) Generalize to the setting where the agent has no dynamics access---Problems (P.2.b) and (P.3.b).
> - (3) Because Problem (P.3.b) is entirely novel, in this work we first focus on the contextual bandits setting.
>
> Importantly, we reiterate that ICB---which instantiates Problem (P.3.b)---is **NOT** subsumed by IRL---which corresponds to Problem (P.3.a). (Of course, one can indeed imagine studying Problem (P.3.b) more generally in the case of arbitrary transition dynamics; but as noted in the manuscript, we leave that to future work).

---

> ### Author Response · Authors · 2021-11-17
> **Response to Reviewer G5nF [Part 3/9]**
>
>
> ---
>
> **(P.3) Inverse Problem**
>
> Now, suppose we---as investigators---are given an observational dataset $\mathcal{D}:=\{x_{1:T},a_{1:T}\}$ of states and actions generated by some agent. The inverse problem deals with determining the true environment parameter $\rho_{\text{env}}$ and/or the agent's belief parameters $\beta_{1:T}$, with respect to which the agent's behavior appears most optimal. Suppose we treat this with a Bayesian formulation:
>
> - **(a) Agent Has Dynamics Access**: In this setting, we seek the following:
>
> $
> \quad\quad \hat{\rho}\_{\text{env}}:=\text{argmax}\_{\rho}\Pr(\mathcal{D}|x\_{t}\sim\mathcal{T}\_{\rho},a\_{t}\sim\text{argmax}\_{\pi}V\_{\text{test}}^{\mathcal{R}\_{\rho}}(\pi))\Pr(\rho)
> $
>
> - **(b) Agent Has No Dynamics Access**: In this setting, we seek the following:
>
> $
> \quad\quad \hat{\rho}\_{\text{env}},\hat{\beta}\_{1:T}:=\text{argmax}\_{\mathcal{\rho},\beta\_{1:T}}\Pr(\mathcal{D}|\beta\_{1},\beta_{t}\sim\text{argmax}\_{f}V\_{\text{train}}^{\mathcal{R}\_{\rho}}(f),x\_{t}\sim\mathcal{T}\_{\rho},a\_{t}\sim\pi\_{\beta\_{t}})\Pr(\rho,\beta\_{1:T})
> $
>
> Note that Problem (P.3.a) is equivalent to a generalization of Bayesian inverse reinforcement learning [39]. More broadly, as the "opposite" to Problem (P.2.a), different flavors of inverse reinforcement learning all share the property that they involve---explicitly or implicitly---the quantity $V_{\text{test}}^{\mathcal{R}_{\rho}}$. For instance, max-margin inverse reinforcement learning [31] seeks $\hat{\rho}\_{\text{env}}:=\text{argmax}\_{\rho}(V\_{\text{test}}^{\mathcal{R}\_{\rho}}(\pi\_{\mathcal{D}})-\text{max}\_{\pi}V\_{\text{test}}^{\mathcal{R}\_{\rho}}(\pi))$.
>
> **On the other hand, Problem (P.3.b) is novel, and has not been studied before**.
>
> Now, as an initial exploration of this problem setting, in this work we consider state transitions that occur independently of past states and actions (which is especially suited for our motivating application to organ allocations).
>
> - In the forward sense, this corresponds to a special case of Problem (P.2.b) that yields the "contextual bandits" setting; this means that decisions can be made greedily: $\text{supp}(\pi\_{\mathcal{R}\_{\rho},\mathcal{T}\_{\rho}}^{*}(x))=\text{argmax}\_{a}\bar{\mathcal{R}}\_{\rho}(x,a)$, with ties broken arbitrarily.
>
> - In the inverse sense, this corresponds to a special case of Problem (P.3.b) that yields the "inverse contextual bandits" setting; this means that $\mathcal{T}$ being unknown is ultimately inconsequential, and we may simply treat $\rho$ as the "reward parameter".
>
> Importantly, inverse contextual bandits is **NOT**---in any shape or form---subsumed by Bayesian inverse reinforcement learning. To be clear, B-IRL is an instantiation of Problem (P.3.a), whereas B-ICB is an instantiation of Problem (P.3.b). The following table summarizes the above discussion:
>
> ||Forward Problem|Inverse Problem|
> |:--|:--|:--|
> |**Agent Has Dynamics Access** (i.e.       |Problem (P.2.a)         |Problem (P.3.a)         |
> |agent optimizes for exploitation only)    |e.g. any optimal control|e.g. Bayesian IRL [39]  |
> |**Agent Has No Dynamics Access** (i.e.    |Problem (P.2.b)         |Problem (P.3.b)         |
> |agent balances exploration + exploitation)|e.g. contextual bandits |e.g. Bayesian ICB (Ours)|
>
> ---
>
> **UPDATE**: We have now included this discussion in a (new) **Appendix B**, which is now appropriately referenced within Section 2. We have also updated some semantics in the (existing) **Section 2** to minimize possible confusion.

---

> ### Author Response · Authors · 2021-11-17
> **Response to Reviewer G5nF [Part 2/9]**
>
>
> ---
>
> **(P.2) Forward Problem**
>
> In the forward direction, an agent seeks to determine the optimal policy that maximizes some notion of expected return. Generally, there are two distinct classes of problems that are of interest: The first assumes the agent has access to environment dynamics (either explicitly, via direct knowledge of $\mathcal{R},\mathcal{T}$; or implicitly, via interaction with the environment in an unassessed training phase). The second assumes the agent has no such access to environment dynamics (therefore learning must occur via interaction with the environment in an assessed training phase).
>
> - **(a) Agent Has Dynamics Access**: In this setting, either the dynamics are explicitly known, so no training phase is required (the agent can simply perform planning using dynamic programming, or simulation-based search, etc.), or the agent may freely interact with the environment in an unassessed training phase. Either way, $V_{\text{train}}^{\mathcal{R}}$ is irrelevant; the agent simply seeks the following:
>
> $
> \quad\quad \pi^{*}:=\text{argmax}\_{\pi}V\_{\text{test}}^{\mathcal{R}}(\pi)
> $
>
> - **(b) Agent Has No Dynamics Access**: In this setting, the training phase is assessed as well---that is, not only does the agent seek the optimal policy for test-time deployment, but they also care about the returns generated during the training phase. On the flip side, note that once $f$ terminates, $V_{\text{test}}^{\mathcal{R}}$ is trivially minimized for any $T$ by setting $\pi=\pi_{\beta_{T}}$. Thus the agent seeks the following:
>
> $
> \quad\quad f^{*}:=\text{argmax}\_{f}V\_{\text{train}}^{\mathcal{R}}(f)
> $
>
> Crucially, in the former, the agent is optimizing $\pi^*$ based on an "exploitation-only" metric. Whereas in the latter, the agent is required to select an $f^*$ that balances "exploitation" and "exploration".

---

> ### Author Response · Authors · 2021-11-17
> **Response to Reviewer G5nF [Part 1/9]**
>
>
> ---
>
> Thank you for your thoughtful comments and suggestions. We give answers to each of the following in turn, as well as pointing out corresponding updates to the revised manuscript:
>
> - Preface on Novelty and Setting
> - (A) Generality of Preliminaries
> - (B) Criteria for Interpretability
> - (C) Relation to Related Work
> - (D) Interpreting the Experiments
> - (E) Algorithm 1 vs. Algorithm 2
>
> We will upload the revised manuscript shortly after this initial response.
>
> ---
>
> ### Preface on Novelty and Setting
>
> Before we begin, kindly allow us to clarify the context of our setting, which should better elucidate the novelty of our problem and solution.
>
> **TL;DR**: In the first instance, our problem setting is actually a *generalization* of the IRL paradigm, in the sense that we attempt to model the trajectory of sequential decision-makers that are *non-stationary*. This is our key contribution, and this has not been done before. Secondarily (and orthogonally), we specifically focus on contextual bandits as a start; however, doing so does not take away from the novelty of this contribution (and---importantly---does not yield a special case of IRL).
>
> ---
>
> **(P.1) Problem Setup**
>
> Consider a Markov decision process $(X,A,\mathcal{R},\mathcal{T})$ as defined in the manuscript. In the most general case, the agent has no access to environment dynamics $\mathcal{R},\mathcal{T}$, so some degree of learning must occur. Let $t$ index the cumulative number of interactions with the environment; if the environment is episodic, then time steps accumulate across resets.
>
> Now, let $f$ denote a probabilistic, online algorithm that learns some parameter $\rho$ of the MDP; for instance, we consider this to be a parameterization of the environment dynamics (but this formalism also applies to learning value functions or black-box policies). And let $\beta_{t}$ denote the agent's $t$-th step knowledge of the quantity being learned; for instance, this would be a parameterization of their probabilistic belief about the environment dynamics. Note that each $\beta$ naturally induces a policy $\pi_{\beta}$:
>
> $
> \pi\_{\beta}(x)[a]:=\mathbb{E}\_{\rho\sim\mathcal{P}\_{\beta}}[\pi\_{\mathcal{R}\_{\rho},\mathcal{T}\_{\rho}}^{*}(x)[a]]
> $
>
> where $\pi\_{\mathcal{R}\_{\rho},\mathcal{T}\_{\rho}}^{*}$ denotes the optimal policy that corresponds to point-valued knowledge that the environment dynamics are exactly $\mathcal{R}\_{\rho},\mathcal{T}\_{\rho}$. Finally, let $T$ be the time step when the algorithm $f$ terminates (e.g. due to some measure of convergence). Broadly, $T$ partitions the agent's behavior into two distinct phases:
>
> - **(a) Training Phase**. This is where $f$ progressively updates the agent's knowledge by interacting with the environment. Within this phase, we may define the *train-time expected return*:
>
> $
> \quad\quad V_{\text{train}}^{\mathcal{R}}(f):=\sum_{t\leq T}\mathbb{E}[r_{t}\sim\mathcal{R}|\beta_{1},\beta_{t}\sim f,x_{t}\sim\mathcal{T},a_{t}\sim\pi_{\beta_{t}}]
> $
>
> - **(b) Testing Phase**. Now after training, the agent ends up with a final $\beta_{T}$, and may then be deployed in the environment. Within this phase, we may define the *test-time expected return*:
>
> $
> \quad\quad V_{\text{test}}^{\mathcal{R}}(\pi):=\sum_{t>T}\mathbb{E}[r_{t}\sim\mathcal{R}|x_{t}\sim\mathcal{T},a_{t}\sim\pi]
> $
>
> Usually, if some knowledge $\beta_{T}$ were learned in a prior training phase, the agent would naturally set $\pi=\pi_{\beta_{T}}$ in the testing phase.

---

> > ### Comment · Reviewer_G5nF · 2021-11-28
> > **Reply to the authors**
> >
> > I would like to thank the authors for their detailed replies to my questions, and to apologize for the delay in my reply. Much of the positioning and goals of the paper are more clear to me after reading their replies to my, and the other reviewers, questions.
> >
> > There are still some points where I'm a bit confused, however.
> >
> > If I understand correctly, you are aiming to solve 2 problems: (i) the inverse RL problem when the transition and reward functions are unknown, and (ii) the inverse RL problem when the RL agent is "non-stationary" -- e.g., changing its objective function over time. Is your claim that _neither_ of these problems has been studied before? It is not clear to me that prior work has never considered these settings. Indeed, point (i) seems to be addressed several times in the survey "A survey of inverse reinforcement learning: Challenges, methods and progress" by Arora and Doshi. For example, section 6.3.1 seems quite related to your objective (i). Additionally, Table 2 in this survey mentions a number of works, e.g. REIRL, CSI, which do not assume that the environment dynamics are known. I think it would be useful to explain how your work relates to these prior works, and perhaps even why prior works assumed knowledge of the transition and reward functions. I found that much of your writing focused on emphasizing the novelty of your setting. I think this point was overemphasized, and it would be more useful to explain _why_ prior work needed these assumptions, and _why_ one should expect that the problem should even be solvable when the reward and transition dynamics are not known, or when the RL agent is non-stationary.
> >
> > It is still not too clear to me why it is necessary to introduce the generality of the MDP setting in your paper, given that you focus on the contextual bandits setting. I think notation and the problem formulation would be more simple and focused if you just mention that prior work (in the general inverse RL framework) has these limitations, and we focus on overcoming these in the simplified setting of contextual bandits. Then you can mention as future work studying the more general IRL problem under your assumptions.
> >
> > ----
> >
> > Aside from these concerns, I think that the authors have addressed many of the concerns raised by the reviewers. However, it seems that addressing these concerns has required quite a large amount of writing. While much of this writing has been added by the authors to the appendix, I think it would be much more useful in the main body of the paper. I think that the paper would be much stronger after a rewrite which incorporates this feedback, and more thoroughly explains the positioning of the paper relative to prior work, the objectives of the work, and the desired features of an interpretable representation and how these features are achieved by the algorithm.
> >
> > For these reasons, I will maintain for now my score of "weak reject" for this paper.

---

> > > ### Author Response · Authors · 2021-11-29
> > > **Re: Reply to authors**
> > >
> > > In inverse problems, there is a difference between *the agent* having access to environment dynamics and *us, as the investigators*, having access to environment dynamics. In our manuscript, when we talk about whether environment dynamics are known or not, we strictly refer to the agent's knowledge.
> > >
> > > When the agent knows (or more generally, has access to) environment dynamics, they can maximize their reward function with a stationary policy (Problem P.2a); if not, they would have to devise a non-stationary policy (that balances exploration and exploitation, Problem P.2b). What has been investigated in the existing literature is the inverse problem when the agent has access to environment dynamics---that is when the agent has an (optimal) stationary policy (Problem P.3a). Although this problem assumes that the agent had access to environment dynamics when planning their policy, it can still be tackled when we, as the investigators, do not have access to the same environment dynamics (this is known as the *batch setting* and the related work you mentioned falls in to this category). The problem we tackle for the first time is the inverse problem when the agent does not have access to environment dynamics---hence their policy has to be a non-stationary policy (Problem P.3b). This problem too can be tackled in the batch setting, and we happen to tackle it in the batch setting, but this is besides our main contribution. While Problem P.3a has been investigated, including in the batch setting where we might not have access to environment dynamics ourselves, Problem P.3b has been never been investigated in any capacity. Note that some of the related work we cite (e.g. [22], [42], [56]) solves Problem P.3a in the batch setting but we did not highlight this aspect simply because our work concerns the agent's knowledge more than the investigator's knowledge of the environment dynamics.
> > >
> > > The reason we introduced the general MDP setting in our paper first is to highlight that Problem P.3b has been highly overlooked by the literature (and formulating Problem P.3b is part of our contributions). Later, we restrict the setting to ICB to offer a first solution. Note that ICB is a special instance of Problem P.3b and having the more general formulation for Problem P.3b helps us articulate the assumptions of ICB much more rigorously (i.e. the fact that $\mathcal{T}(x,a)=\mathcal{T}'$ for all $x,a$ for some $\mathcal{T}'$).

---

> ### Author Response · Authors · 2021-11-21
> **Dear Reviewer G5nF**
>
> Once again, thank you for your invaluable feedback. We were wondering whether our response and the revised manuscript addressed your concerns. If you have any additional comments, please let us know, we would be happy to address them.

---

> > ### Author Response · Authors · 2021-11-27
> > **Dear Reviewer G5nF**
> >
> > Thank you very much for your helpful comments. We are sorry to disturb you again but we were just wondering whether our response has adequately addressed your concerns. If not, please let us know what remaining concerns you have; we are very eager to respond to those as well.

---

### Author Response · Authors · 2021-11-19
**Dear Reviewers**

Thank you again for your invaluable feedback. We have now uploaded the revised version of our manuscript. Here is a summary of the changes we have made (we highlighted all changes in *blue* in the manuscript):

- We replaced Figure 1 with a brand **new figure**, which relates more directly to the results we present in Figure 4. We hope that this new figure better motivates the need for a method that is capable of capturing (and describing) non-stationary decision-making behavior.

- We updated some semantics in Section 2 to make the exposition of our setting clearer. In particular, we now denote the true environment parameter with $\rho_{\text{env}}$ instead of $\rho^*$ and we denote the agent’s policy at time step $t$ (for the case with no access to environment dynamics) with $\pi\_{\beta\_t}(x)[a]:=\mathbb{E}\_{\rho\sim\mathcal{P}\_{\beta\_t}}[\pi^*\_{\mathcal{T}\_{\rho},\mathcal{R}\_{\rho}}(x)[a]]$ such that $a\_t\sim\pi\_{\beta\_t}(x\_t)$.

- We added a **new Appendix A**, which provides further discussion on experiments. It explains why exactly the existing benchmarks fail or do not perform as well as B-ICB/NB-CB does in our setting. It also includes **new experimental results** in a semi-synthetic environment, where the behavior changes suddenly, more than once.

- We added a **new Appendix B**, which clarifies how conventional formulations of reinforcement learning (RL), inverse reinforcement learning (IRL), and contextual bandits (CB) relate to our novel formalism, that is the inverse contextual bandits problem. There, we define mathematically what “having unrestricted access to dynamics” means, and precisely compare and contrast problems in sequential decision-making into four classes: Forward problems with access to dynamics (Problem 1A), forward problems with no access to dynamics (Problem 1B), inverse problems with access to dynamics (Problem 2A), and inverse problems with no access to dynamics (Problem 2B). Then, RL is a particular instance of Problem 1A, IRL of Problem 2A, and CB of Problem 1B. Importantly, this precisely identifies ICB as the first instance of Problem 2B ever considered.

- We added a **new Appendix C**, which discusses interpretability of our results and technical aspects of our algorithms. In particular, it defines our concrete criteria for interpretability that we desire of any solution to ICB. It also discusses how to interpret the results we present in Section 5, how Algorithm 1 and 2 (B-ICB and NB-ICB) provide different interpretations that could be beneficial for different use cases, and how ICB relates to existing frameworks for analyzing sequential decision-making (such as IO-HMMs and bandit setups).

- We also uploaded the necessary code to replicate our experimental results as supplementary material.

---

### Decision · Program_Chairs · 2022-01-20

**Decision:**

Reject

**Comment:**

Summary: This paper studies an inverse (linear) contextual bandits (ICB) problem, where, given a $T$-round realization of a bandit policy’s actions and observed rewards, the goal is to design an algorithm to estimate the underlying environment parameter, along with the “belief trajectory” of the bandit policy. A particular emphasis is placed on the belief trajectory being “interpretable” and capturing changes in the policy’s “knowledge of the world” over time.

The paper’s main contributions are (i) formalizing the inverse contextual bandits problem, (ii) designing two algorithms for this problem based on two different ways of modelling beliefs of the bandit policy, and (iii) providing empirical illustrations of how their algorithm can be used to investigate and explain changes in medical decision-making over time

Discussion: This paper has received high quality, long and detailed reviews that highlighted some flaws, in particular in the well-posedness of the problem and the clarity of the writing. The authors' response was long and detailed as well, and its quality was recognized by the committee.
However, the consensus is that this work would require a full pass allowing to include most of the feedback received in the main text rather than in appendices, to discuss related problems in the literature in more depth and perhaps to refocus the exposition on the problem considered.

Recommendation: Reject.